**communications** sustainability

# The river Rhine transports around 4,000 tonnes of macrolitter towards the North Sea each year

Nina Gnann [1,6], Katharina Höreth [2,6], Nicolas Schweigert [3], Mariele Evers[2], Thomas A. Ternes[4] & Leandra Hamann [5] ✉

Improper waste disposal has led to a rising accumulation of litter in aquatic environments, but long-term surveys of riverine macrolitter particles are rare. Here we report an analysis of continuous in-river monitoring over 16 months, using a custom-made physical-interception litter trap in the Rhine River, one of Europe's largest rivers. In collaboration with citizen scientists, we collected 20,339 floating macrolitter items larger than 1 cm that we classified according to international standards. We find that private consumers are the largest source of macrolitter at 56.4%. Plastics comprised 69.7% of macrolitter by particle number but only 14.8% by weight, underscoring the relevance of non-plastic materials. Based on our data, we estimate the annual macrolitter transport towards the North Sea to amount to 27 to 42.2 million items, or 3,010.5 to 4,707.5 t in the Rhine River, which significantly exceeds previous estimates. Macrolitter abundance fluctuated by a factor of 41 between biweekly samples and was higher at times of rising discharge. As such, we suggest that long-term monitoring is essential, and river discharge forecasts could serve as a predictor to guide clean-up efforts.

Improper waste disposal has led to a rising accumulation of litter, particularly plastic litter[1–3], in aquatic environments[4,5]. If current trends persist, the global quantity of plastic litter in rivers, lakes, and seas, either as micro- (<5 mm), meso- (5 mm - 25 mm), or macroplastics (>25 mm), is projected to triple to more than 493 Mt in 2060[6]. This increase is also expected to amplify associated adverse effects such as entanglement and ingestion by organisms, a rise in adverse effects on human health, and higher flood risks due to clogged ML drainage systems[7,8]. Current measures to reduce macrolitter (ML), which includes any anthropogenic material >25 mm, include reducing the consumption of products prone to becoming ML, improving recycling rates, conducting clean-ups, and confining leakage pathways[7]. To prevent ML leakage into the environment through reduction measures, it is essential to monitor ML and identify its sources, drivers, and quantities, and assess the effectiveness of the implemented measures[1,9].

Rivers play a key role in ML transport, making them effective sites for ML monitoring closer to its origin on land[10,11]. Riverine monitoring methods include observational methods for floating ML, e.g., visual human observation or remote sensing, and physical-interception methods, e.g., net sampling, boom structure sampling, waste collection activity, or sediment sampling[10,12]. ML monitoring in rivers provides item or mass counts per volume or time, ML profiles along the river cross- or longitudinal section, and extrapolations for annual emissions into the sea[13–15]. For example, based on the visual human observation method, it was estimated that European rivers transport between 307 and 925 million ML items per year into the sea[13]. However, observational methods are prone to errors due to visibility restrictions, field conditions, human error, short observation time, and lack of validation options[10]. The physical-interception methods are more reliable but invasive, resource-intensive, and time-consuming. For this reason, continuous, large-scale and long-term data for floating and suspended ML is still scarce, especially in large navigable rivers[13].

In this study, we continuously monitored ML by using a custom-made litter trap (LT) as an interception-based method for 16 months at a fixed location in the Rhine River. The Rhine is one of Europe's largest navigable rivers, which connects nine European countries and counts 49 million people in its basin[16] (Fig. 1A). The stationary LT catches floating and submerged ML up to 80 cm below the surface and is cleaned biweekly by volunteers as part of a citizen science project (Fig. 1B). The sampled ML was recorded based on the 183 categories of the Joint List of Litter Categories for Marine Macrolitter Monitoring (MSFD list)[17] to comply with harmonisation efforts and reach international monitoring standards that are required

[1]Department of Geosciences, Eberhard Karls University Tübingen, Tübingen, Germany. [2]Department of Geography, University of Bonn, Bonn, Germany. [3]K.R.A.K.E. (Kölner Rhein-Aufräum-Kommando-Einheit) e.V., Cologne, Germany. [4]Federal Institute of Hydrology, Koblenz, Germany. [5]Bonn Institute for Organismic Biology, Section 2, Animal Diversity, University of Bonn, Bonn, Germany. [6]These authors contributed equally: Nina Gnann, Katharina Höreth. ✉e-mail: hamannleandra@gmail.com

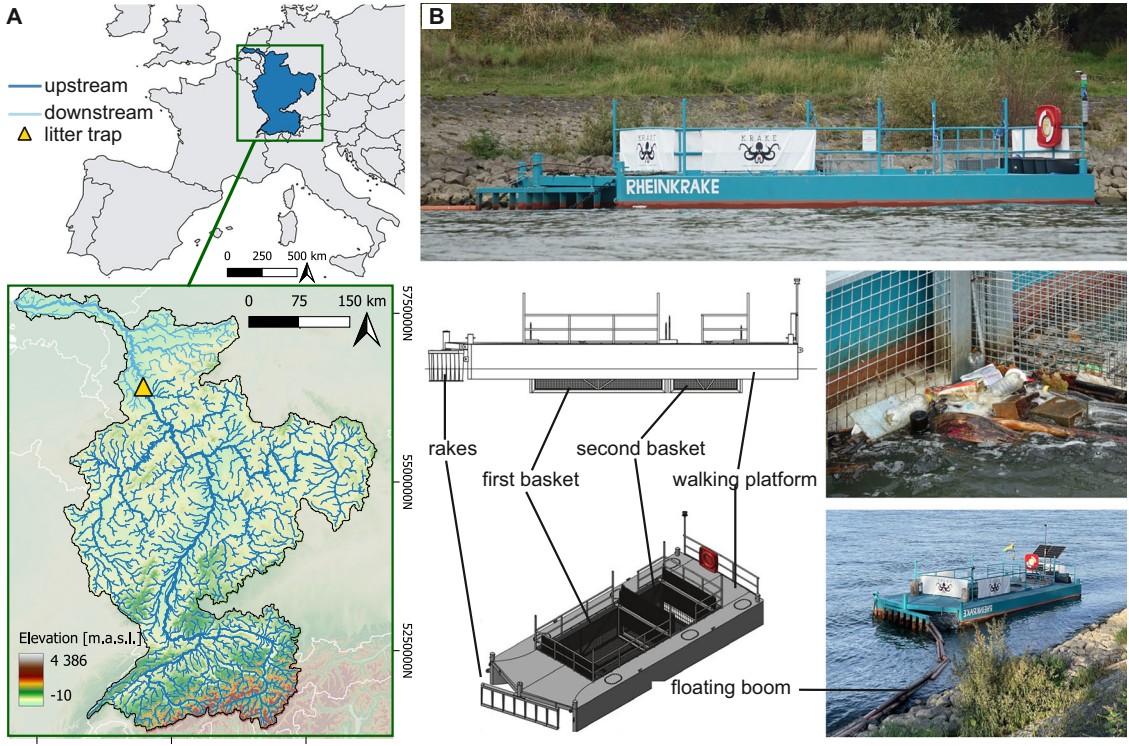

**Fig. 1 | Location and setup of the litter trap (LT). A** Location of the LT (yellow triangle) at Cologne within the Rhine basin (blue), which divides into an upstream (dark blue) and a downstream (light blue) basin area[68–70]. **B** Design drawing (Design credits: **A** Nebel/LUX-Werft) and photos (Photo credits: Leandra Hamann) of the custom-built LT with entrance on the left side. ML is retained inside the two baskets. The floating boom connects to shore and guides ML towards the LT (Supplementary Fig. 2).

to achieve the objectives of the EU Marine Strategy Framework Directive (2008/56/EC). We extended the MSFD list by usage, source, and consumer sector groups to identify entry paths and link those to environmental and cultural factors as potential drivers of ML in the Rhine. Based on a subset of one year, which covered no extreme flooding events, we extrapolate the annual ML transport towards the North Sea in a bottom-up, data-driven manner.

## Results

### ML consists primarily of small, single-use plastic products from private consumers

The LT collected 20,339 ML items (24.09.2022–13.01.2024), of which 17,523 were collected during the statistical year (19.11.2022–18.11.2023). ML items without weight artefacts, such as filled water bottles or soaked clothes, weighed 1,142.3 kg (10,238 #) in the statistical year. This equals an average weight of 111.6 g per ML item across all materials. Plastic ML items weigh on average 24.3 g. Extrapolated to all ML in the statistical year, this equals 1,955.2 kg year$^{-1}$ for all materials and 289.6 kg year$^{-1}$ for only plastics. In the full sampling period, plastics have the highest material share by item number with 69.7%, followed by worked wood (14.9%), glass & ceramics (5.7%), paper & cardboard (2.6%), metal (2.6%), rubber (1.6%), chemicals (1.6%), food waste (0.8%), and cotton & textiles (0.5%, Fig. 2A). From 183 categories, 145 of the original MSFD list were found during the whole sampling period. The Top-15 categories account for 74.3% of ML (Fig. 2B), with the top-5 categories being small foamed polystyrene fragments (20%, 4077 #), wooden remains of fireworks (9.7%, 1964 #), fragments of non-foamed plastic (9%, 1823 #), glass bottles (5.3%, 1078 #), plastic caps/lids drinks (4.5%, 921#). Most ML with 71% is <20 cm (Fig. 2C). ML <5 cm represent 23% but was not systematically sampled due to the LT mesh size. During the monitoring process, ML of one category but with different sizes were often photographed together. Therefore, ML of mixed size accounts for 61.4%.

Single-use products cover 64 categories and have a share of 40.4% (8210 #) of all ML (Fig. 2D). Single-use plastic ML accounts for 23.7% (4810 #) of all ML. The most common single-use ML is wooden remains of fireworks, followed by plastic caps/lids drinks, both also in the top-15 ML items, and plastic crisp packets/sweets wrappers with 4.5% (906 #) of all ML. ML with multi-use has a share of 7.6% (1537 #) of all ML. Multi-use plastic items make up 1.9% (383 #) of all ML. ML categories which include potentially single- and multi-use items such as glass bottles, flower pots, and plastic toys are assigned as ambiguous and accounts for 21.4% (4360 #). Non-identifiable ML has a share of 30.6% including all fragments (Fig. 2D).

Based on the 13 use groups in the MSFD list, ML with unidentified use accounts for 54%, followed by food-consumption-related ML with 19.9%, recreational-related ML with 12.8%, and smoking-related ML with 6.5% (Fig. 2E). The adjusted classification shows that 99.6% of ML is land-based, whereas water-based ML represents 0.4% (Fig. 2F). The water-based ML is mainly related to fishery. Of the land-based ML, private consumers have the highest share with 56.4%, followed by industrial with 5.9%, and traffic, infrastructure, and building with 1.6%. The other 36.0% could not be assigned to any specific type of origin and contained mainly fragments (Fig. 2F). Of the private consumer items, drink-, fireworks-, food-, and smoking-related items is the most common ML with 18.1%, 10.7%, 9.9%, and 6.5%, respectively. Packaging amounts to 76% of the industry-related ML, which equals 4.5% of all ML. When all items related to packaging are combined (40 categories) regardless of any levels, they amount to 32.2% of all ML. Lids (6 categories) add up to 7.2% of all ML.

### ML abundance over one year and the influence of environmental factors

During the statistical year, ML per biweekly sampling periods varies by a factor of 41 from a minimum of 67 items (21.10.2023) to a maximum of 2724 items (14.01.2023) (Fig. 3A). The latter date is the first sampling after New Year's Eve 2022/2023. At this sampling date, a total of 1943 firework-

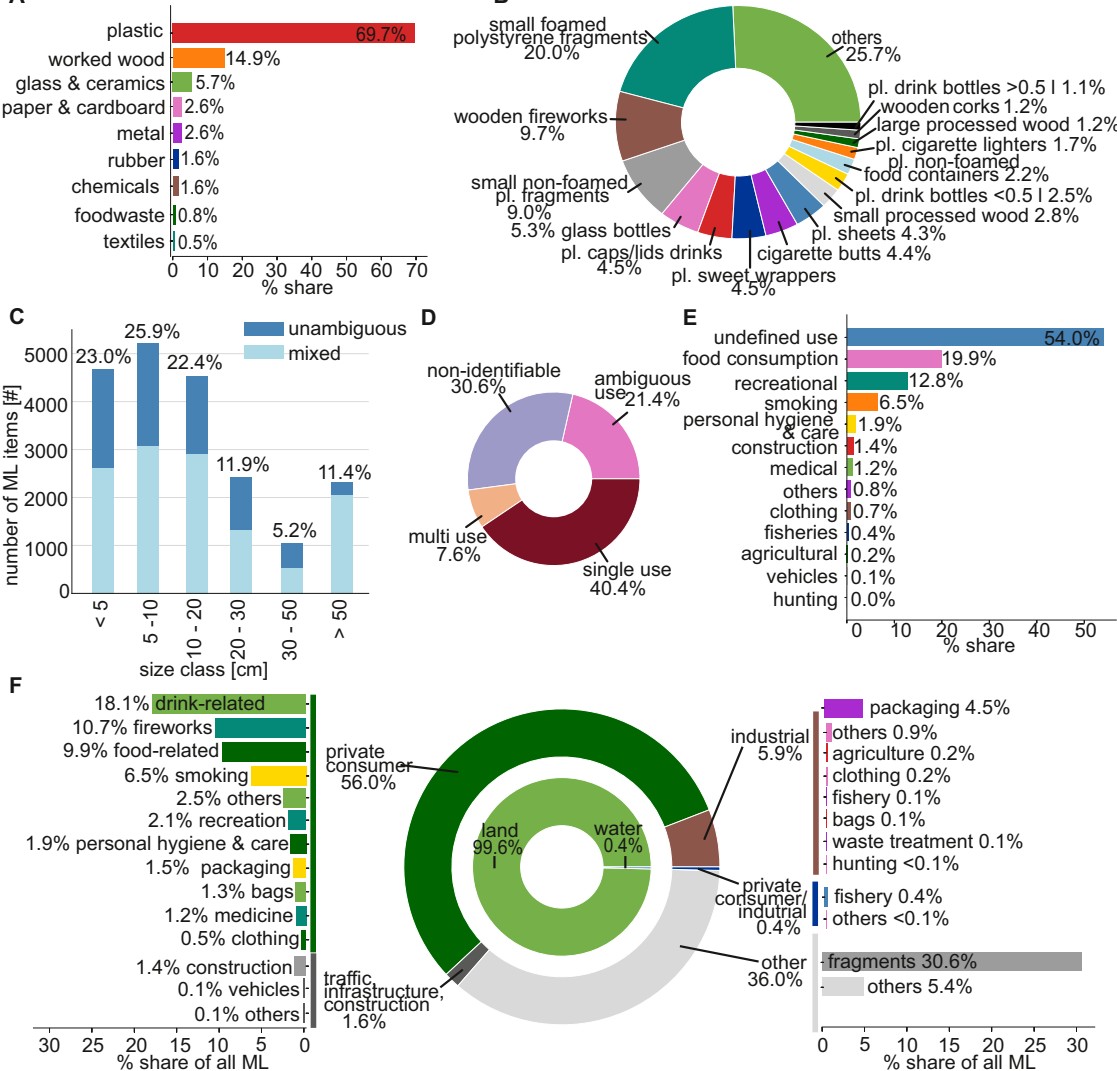

**Fig. 2 | Analysis of ML items by number (20,339 #) retained by the litter trap (LT) during the sampling period of 16 months (24.09.2022–13.01.2024). A** Material composition of ML. **B** Top 15 ML categories. **C** Number of ML and percentage share according to six size classes. The unambiguous class corresponds to objects of the assigned size class, while the mixed size class contains objects of this size class and any smaller size. **D** Share of ML based on use durability. **E** ML usage grouped according to the ML list prepared by the MSFD Technical Group on Marine Litter (MSFD TG ML) in close collaboration with EU Member States and the Regional Sea Convention[17]. **F** ML grouped according to the adjusted classification developed in this study, including three levels describing source, sectors, and usage.

related ML was found, mainly consisting of single-use wooden fireworks (Fig. 3B + C). This is about three times as much as the mean across all sampling dates, which is 674 ML items. For the second New Year's Eve, 2023/2024, 59 firework-related ML items were found, which is 3% of the previous year. Except for New Year's Eve, we could not identify a relation between a cultural event and ML abundance, material type, or usage (Fig. 3C).

Plastic ML has the highest share of all materials throughout the year, except for the sampling date after New Year's Eve 2022/2023 (Fig. 3A). The most common identifiable use type is single-use with an average of 36.8% (range 6.7–83%), followed by ambiguous use with 29% (range 9.4–53.8%), non-identifiable use with 25.6% (range 0–80.4%), and multi-use with 8.6% (range 1.8–21.9%, Fig. 3B). The high proportion of non-identifiable use of ML is due to a high number of fragments (Fig. 3C). Fragments show a large variation in ML share, ranging from 0 to 83.7% (0–899 #) per sampling date over the total sampling period. Drink-, food-, packaging- and smoking-related ML was also found frequently throughout the full sampling period and ranged from 2.4 to 44.7% (12–358 #), 2.6 to 36.2% (3–180 #), 3 to 159 # or 1.4 to 32.5% (3–159 #), and 0 to 16.8% (0–399 #), respectively (Fig. 3C).

Of the environmental factors, rising discharge shows a strong positive correlation with the number of ML items and number of ML categories ($\rho = 0.83$ and $0.81$, $p < 0.05$, Fig. 4). Three outliers were found in each correlation. Their respective sampling dates could not be linked to cultural events, other than New Year's Eve, or any environmental factors. The smaller the object, the stronger the correlation with the rising discharge component (Supplementary Table 2). For example, objects with a size of 1 to 5 cm have a $\rho$-value of 0.83, while the most significant size class only shows a correlation coefficient of 0.55. Wind speed and precipitation moderately correlate to the number of ML items ($0.51 < \rho < 0.57$) and number of ML categories ($0.48 < \rho < 0.51$, Supplementary Table 2). Among the materials, worked woods, plastics, metals, and rubber positively correlate with rising discharge ($\rho > 0.78$). All other materials, except organic food waste, show a moderate positive correlation with rising discharge ($0.51 < \rho < 0.62$). All materials show a positive correlation with either one or all of discharge, precipitation, and wind speed ($0.40 < \rho < 0.75$, Supplementary Table 2). Of the 145 found categories, only 12 categories show a strong positive correlation with rising discharge ($\rho > 0.70$). This includes categories such as foamed and non-foamed fragments, wooden corks, small plastic drink bottles, plastic lids, and plastic cigarette lighters (Supplementary Table 2).

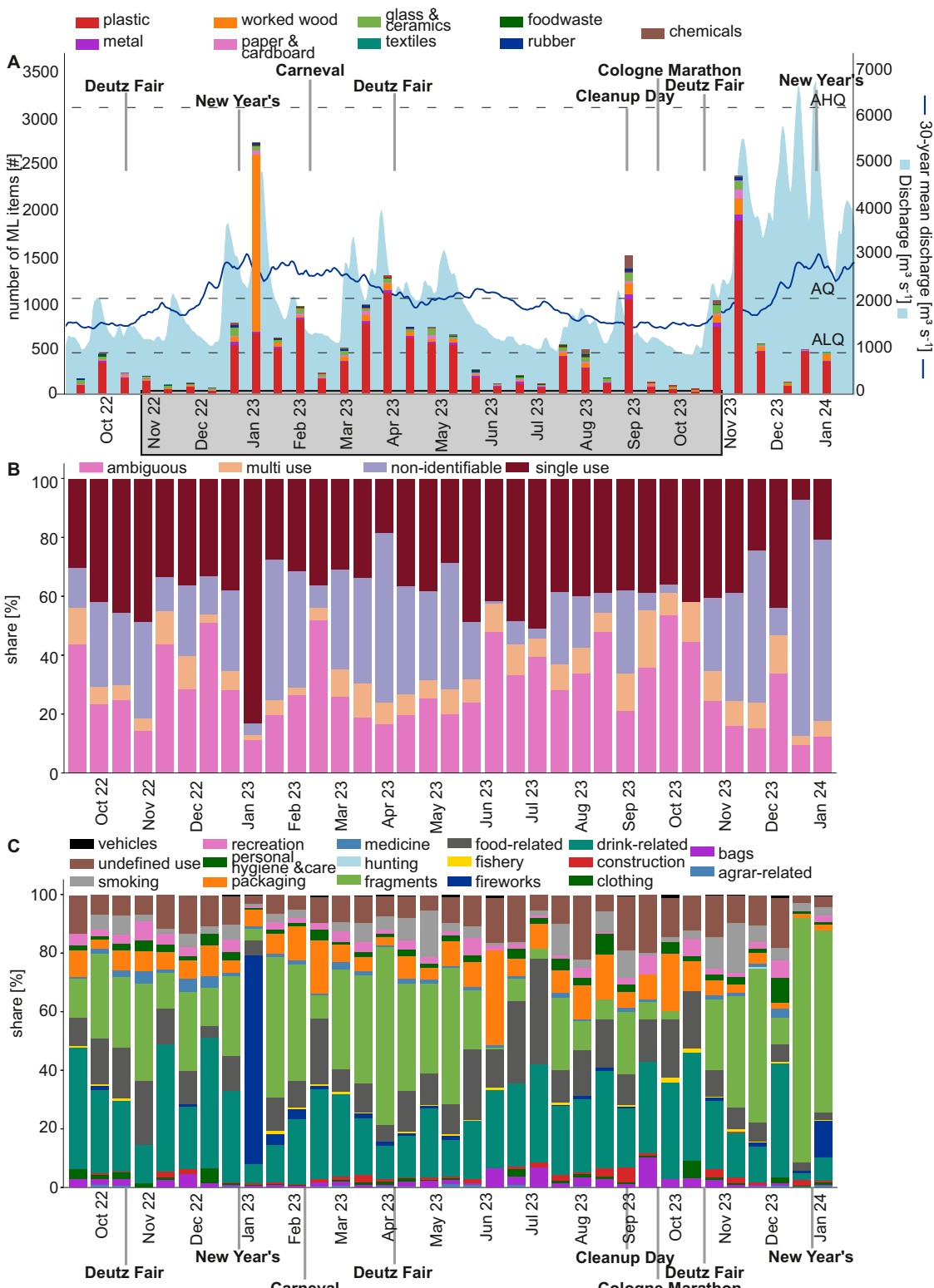

**Fig. 3 | ML quantities over 16 months for each sampling date with relation to discharge and cultural events. A** ML composition by material and discharge (light blue)[53]. The dark blue line represents the mean discharge level at the Cologne gauging station from 30 years (1993 until 2022)[52]. The dotted lines represent average high discharge (AHQ), average discharge (AQ), and average low discharge (ALQ).

Please note that the floating boom was detached before 19.11.2022 and after 18.11.2023 (statistical year). The light grey box at the x-axis indicates the statistical year. **B** ML composition based on use type for each sampling date. **C** ML composition based on sources for each sampling date.

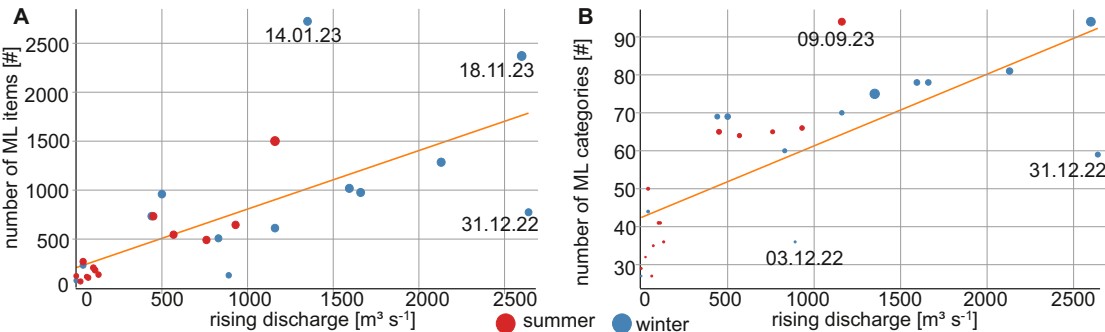

**Fig. 4 | Relation of ML to rising discharge. A** Number of ML items (#) to rising discharge ($\rho = 0.83$, $p < 0.05$). **B** Number of ML categories to rising discharge ($\rho = 0.81$, $p < 0.05$) for each sampling date (35 sampling dates, 20,339 ML items). Red dots represent summer sampling dates, and black dots show winter sampling dates. Outliers are marked with the respective sampling date.

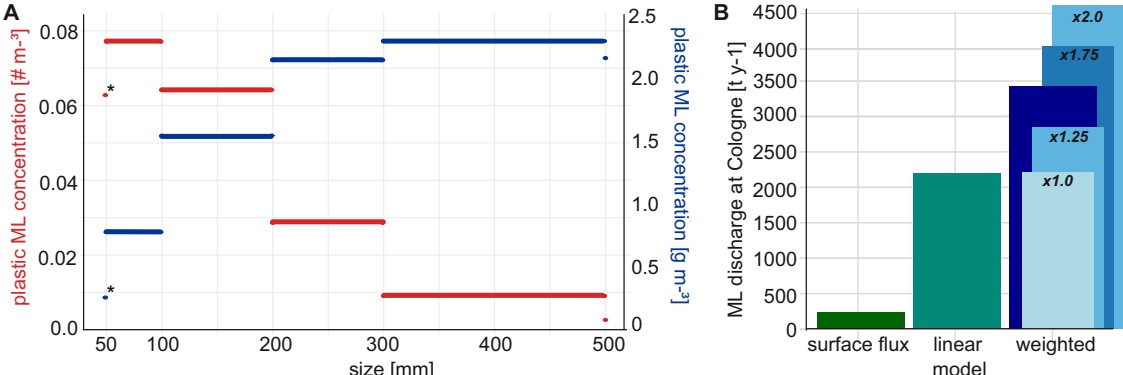

**Fig. 5 | Plastic ML distribution in the Litter Trap (LT) and total ML discharge at Cologne gauging station with different extrapolation scenarios. A** ML plastic concentration based on number m⁻³ (red) and mass m⁻³ (blue) for every size class. The asterisk (*) indicates that this size class is smaller than the systematic sampling size and the calculated concentration for <50 mm is likely underestimated. **B** ML discharge at Cologne based on the three different extrapolation models. The weighted extrapolation model was calculated with a vertical distribution factor ranging from 1 to 2 in increments of 0.25 (light blue bars) around the base value of 1.5 (dark blue bar).

Small wooden objects have a strong positive correlation with precipitation ($\rho > 0.70$), and facemasks negatively correlate with temperature ($\rho > -0.76$, Supplementary Table 2). ML abundance during winter months (November-April) is significantly higher than in summer months (May-October, Mann-Whitney-U: $p = 0.02$, $r = 0.45$, $n = 15/11$, Fig. 4). There is no significant difference between the four seasons autumn, winter, spring, and summer (Mann-Whitney-U: $p = 0.13–0.95$, $r = 0.04–0.46$, $n = 6–7$).

### The Rhine at Cologne transports around 53,000 ML items every day

The LT filters 127,617.6 m³ day⁻¹ of water, which equals around 0.08% of the mean daily discharge of the Rhine at the Cologne gauging station. On average, 48 ML items and 5.4 kg of ML are caught in the LT daily, which equals a concentration of 0.00036 ML items m⁻³ and 40.1 mg m⁻³. The concentration of plastic ML is 0.00024 ML items m⁻³ and 5.9 mg m⁻³. The number of plastic ML per unit volume increases with decreasing size, whereas plastic ML mass concentration increases with increasing size (Fig. 5A). The ML surface flux in the LT is 202.6 tonnes and 1,815,666.8 ML items for the total Rhine's width per year (Fig. 5B). This equals 207.3 ML hour⁻¹ at Cologne. The linear ML discharge estimate at Cologne results in 53,261.2 ML items day⁻¹. This equals 5.9 t day⁻¹ or 2,169.0 t year⁻¹. When including a vertical distribution factor of 1.5 and a rising discharge factor of 1.0425, the weighted ML discharge estimate is 9.3 t day⁻¹ and 3,391.8 t year⁻¹ at Cologne (Fig. 5B). The total ML discharge towards the North Sea is 3,010.5 t year⁻¹ for the linear ML discharge estimate and 4707.5 t year⁻¹ for the weighted ML discharge estimate.

## Discussion

Our study estimates that the Rhine transports 27.0 to 42.2 million ML items into the North Sea annually, equivalent to 3,010.5 to 4,707.5 t of ML by weight. Previous estimates of ML are at 0.5-3.5 t y⁻¹ [18], 3-5.4 t y⁻¹ [11], 9.1-31.8 t y⁻¹ [19], and 20-31 t y⁻¹ [20], but exclusively focused on plastics. This is 22 to 286 times less than our findings with 445.9 to 697.3 t y⁻¹ of plastic ML, a discrepancy primarily attributed to methodological differences. Our study determined an average plastic weight per item of 24.3 g, compared to 5.38 g in the previous estimates for the Rhine [11,18]. This could be due to measuring wet weight and the type of scales used for weight measurements with a minimum weight of 1 g, limiting the accuracy of small and lightweight items. This was partly circumvented by measuring several small items of the same category together and then dividing the weight by their number. Yet, with an average weight of 111.6 g per item for all materials, our results lay within weights reported in beach ML monitoring studies where plastic ML averaged 15.6 g per item, and other materials ranged from rubber with 76.8 g to wood with 154.7 g [21]. Another key factor influencing our higher estimates is our long and continuous sampling period of 52 weeks. In the previous plastic ML extrapolations for the Rhine, van der Wal et al. (2015) [20] based their estimate on two sampling dates, and Vriend et al. (2020) [18] calculated the average weight based on 508 items and scaled up the total plastic ML transport in combination with data from visual observation of 15 hours [18]. Our results, covering a total sampling time of 11,808 hours, show that ML abundance can vary by a factor of 41 between bi-weekly sampling dates. ML abundance may fluctuate even more during shorter sampling periods.

Hence, limited sampling periods may not represent the annual ML load in rivers, leading to insufficient data for extrapolation models. More data needs also to be gained for the horizontal and vertical distribution of ML in rivers, that are influenced by ML density, wind, river bends, currents, or shipping. The sensitivity analysis of the vertical distribution factors shows that the extrapolation model is quite sensitive to this factor (Fig. 5B), which leads to potential under- or overestimates of ML along the vertical dimension. For example, a change of vertical distribution factor from 1 to 2 duplicates the ML discharge. In another study on the estuaries of three other rivers in Germany, Ems, Weser, and Elbe, the number of ML items increased by a factor of 1.3 up to 292.4 from the surface to the water column depending on the river and ambient conditions[22]. Regarding the size distribution, the total estimates are probably underestimating ML under 5 cm as 3.818 cm is the lower limit of the systematic sampling size. Small ML below this size limit, such as fragments, cigarette butts, bottle caps, lids or toys, have a share of 23% of all ML by number (Fig. 2C). Based on our finding that smaller ML is higher in number and lower in weight (Fig. 5A), this underestimate of ML will more effect the extrapolation of total ML number than total ML weight transported in the Rhine.

Variation of extrapolated ML transport is a common problem and was found to vary by up to five orders of magnitude due to errors in item-to-mass conversion, missing calibration and validation, the under-representation of river dynamics, and the influence of terminal sinks for ML[23,24]. Our results show that extrapolated ML can vary by a larger factor than five, mainly due to non-continuous sampling, which is crucial considering rivers are inherently lotic environments. According to the estimated ML transport based on our long-term monitoring, the Rhine is a major contributor to European ML emissions into the oceans, estimated at 307 and 925 million ML items per year[13]. This can also be seen based on the calculated ML surface flux of the Rhine with 207.3 items h⁻¹, which is in a similar range as other large European rivers such as the Danube, Vistula, or Adour[13]. Compared to the Yangtze River in China, globally the number one river in transporting ML from land to sea with 333,000 t year⁻¹ [25], the Rhine is contributing around 73.5 times less ML to global marine pollution with a maximum estimated ML transport of 4,708.0 t year⁻¹. However, when assuming that littering is connected to human activity[26,27] and ML transport is set in relation to river catchment population[16], the Rhine turns out to only transport 7.6 times less ML per person (0.091 kg y⁻¹ per person, 49,596,235 people) than the Yangtze river (0.69 kg y⁻¹ per person, 480,679,188 people). Relatively speaking, the Rhine contributes substantially more to global ML pollution than previously estimated. However, these comparisons are limited by the fact that previous estimates of total ML transport in other rivers were based on non-continuous sampling methods. These need to be reviewed and potentially redone in the future to allow comparison on harmonised sampling methods and determine a more precise contribution of each river to global ML transport into the sea.

Plastics are often the focus of studies because they have adverse effects, such as harm to aquatic organisms, chemical contamination, and economic losses[7], and are the most abundant material in riverine systems. For example, the plastic share was 50.5% in Llobregat and 67.7% in Besòs River, Spain[2], 77% in Rhone Delta, France[25], 91% to 93% in Ross River, Australia[28], and 97% in Orange-Vaal River, South Africa[29]. With a share of 69.7% by number, which presumably underestimates the share of small ML due to the mesh size of 3.818 cm, we also found plastics to be the most common material in the Rhine. However, plastics account for only 15.2% for the while sampling period (14.8% for the statistical year) of the weight, leaving the other eight materials with a share of 84.8%. These other material groups and their additives might pose a threat to the environment, too. For instance, synthetic polymers from textiles such as polyester, polyamide, and poly-olefins make up more than 60% of the global fibre consumption, and fibres are the most common microplastic particle shape in the Rhine[30] and other environmental compartments[31]. Ingestion of microplastic fibres and particles by freshwater organisms can result in reduced feeding and reproductive activity, blocking off the digestive tract and even causing death[32].

Additionally, textiles may contain toxic substances from manufacturing processes, e.g., Benzothiazole, azo dyes, and perfluorinated chemicals[31]. Rubber products such as tires contain zinc, other heavy metals, and other compounds such as phthalates that can leach into the environment and cause toxic effects in organisms and humans[33]. Even natural, less persistent materials, such as worked wood, paper, cardboard, and food wastes, may contain toxic or harmful additives. This includes bisphenol A (BPA), an endocrine-disrupting chemical used in cardboard and paper coating[34], and Per- and Polyfluoroakyl substances (PFAS), which are common in impregnated paper for food packaging and coated woods and are known as "forever chemicals" because of their extraordinary persistence in the environment with potential ecotoxicological effects[35]. Microcrystalline waxes are petroleum-based paraffins used for the surface treatment of fruits, e.g., mango and avocado. Even though they do not pose a safety concern during consumption in the currently allowed dosage[36], petroleum waxes are an emerging marine pollutant that can potentially threaten aquatic life[37]. Inorganic materials like glass, ceramics, and metals do not decompose naturally in the environment. They can pose a physical threat through sharp edges[38] and release hazardous substances like heavy metals during corrosion or fragmentation[32]. Reactive agents from medication, drugs, and their breakdown products may pose additional stress in soil and aquatic environments[39]. We conclude that ML abundance and potential environmental threats could be highly underestimated when only focusing on plastics instead of considering all ML material types.

To reduce ML input into rivers and avoid their associated adverse effects, our monitoring of ML sources, categories, and drivers aids in developing targeted reduction measures. The source-based classifications of ML categories developed in this study allowed us to identify the sources of 64% of ML (Fig. 2E). This is higher than the 46% identified using the original MSFD groups (Fig. 2D) and also higher compared to previous studies, with 53.5% on marine beaches in Portugal with a classification using seven source groups[40] and 38.8% at the riverbanks of Erzen River, Albania, using eight source groups[41]. One of the primary sources is single-use items. Single-use items have a total share of 40.6% of all ML, according to our classification system. Of those, 98% come from private consumers, which corresponds to a total share of 39.9%. This includes tobacco products with filters with a share of 4.4% and fireworks-related items with a share of 10.7% of all ML, both being unregulated in the EU so far even though they contain environmental pollutants like toxic residues or might cause light, noise, or air pollution[42,43]. A cross-comparison of use durability and private consumers shows that only 1.8% of all ML is multi-use ML from private consumers. Thus, single-use ML, especially from private consumers, is a key source group where targeted measures can substantially impact ML reduction.

Closing the ML leakage of the Top-15 categories into the Rhine could reduce 74.3% of ML (Fig. 2B). This includes e.g., wooden remains of fireworks, plastic sweet wrappers, cigarette butts, plastic sheets, plastic cigarette lighters, and large processed wood, which are in the EU unregulated items, besides the general regulations regarding waste disposal. Hence, there is potential to further reduce riverine ML through measures that incentivize proper waste disposal and ban certain products or substitute harmful materials. Surprisingly, the Top-15 categories also contain items, i.e., glass bottles, plastic caps/lids drinks, plastic sweet wrappers, plastic drink bottles, and non-foamed food containers, which are already targeted in EU regulations. For example, glass bottles with 5.3% (1078 #) of ML have the largest share of drink-related ML, even though they either fall under the deposit system to be reused or have to be disposed of in designated glass containers in Germany since 2005[44]. One option to further decrease glass bottles in the Rhine could be to extend the deposit system to spirits, wine and sparkling wine bottles, which currently are not subject to a deposit.

As there are no benchmarks regarding ML pollution available for the Rhine from previous studies, it is challenging to evaluate the effectiveness of these reduction measures. As of now, only a comparison to other countries with no, less, or more reduction measures might indicate some effectiveness. ML categories subject to previous regulations include lightweight plastic carrier bags with 0.3% (EU Directive 215/720) and single-use items such as

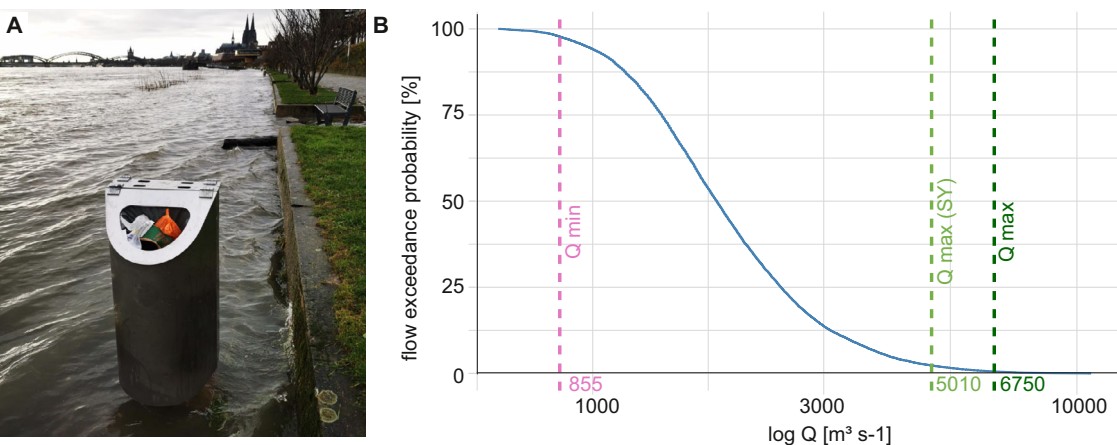

**Fig. 6 | Water discharge in the Rhine at the Cologne gauging station. A** Flooded litter bin at a water level of 553 cm and a discharge of 4760 m³ s⁻¹ (Photo credits: Frank Fuchs/KRAKE e.V.). **B** Flow duration curve of the Rhine based on medium daily discharge data at the Cologne gauging station between the years 1993 and 2022[52]. The dashed line in pink indicates the discharge range of the total sampling period and the statistical year's lowest water discharge during the sampling period at 97.75% exceedance (855 m³ s⁻¹), the light green line indicates the highest discharge of the statistical analysis at 2.32% exceedance (5010 m³ s⁻¹) and the dark green line shows the highest of the whole sampling period at 0.57% exceedance (6750 m³ s⁻¹) (dark green).

plastic straws with less than 0.1%, and plastic cotton bud sticks with 0.3% (EU Directive 2019/904). In comparison to other regions, plastic bags were sampled with 9.7% in the Thames Estuary[45], plastic straws made the Top-3 in waste collection activities in the Ross River in Townsville, North Queensland, Australia[28], and plastic cotton bud sticks show a share of 5.3% in beach samples in Italy before 2021[46]. Recently introduced regulations offer a great opportunity to evaluate their effectiveness. This includes the EU regulation on plastic caps/lids in which all lids of single-use drinking containers up to 3 litres must be fitted with a tethered cap from July 2024 onwards (Directive (EU) 2023/1060). Based on our data, the regulation can potentially reduce 4.5% of floating ML in the Rhine. If all six lid categories were considered, including wooden corks and metal lids, the EU regulation could reduce ML by 7.3% (1473 #). In case this regulation leads to a substantial reduction of specific ML items, further regulations or bans could be supported by these findings for example for fireworks related ML.

Another challenge with reducing the Top-15 categories is that some of the most contributing categories are ML fragments, such as foamed plastics, non-foamed plastics, and plastic sheets, which account for 20%, 9%, and 4.3% of all ML, respectively. We assume that the share of fragments is likely underestimated because fragments are typically smaller than non-fragmented ML items (median fragment size: 5 to 10 cm, median non-fragment size: 10 to 19.9 cm), and the LT does not systematically retain litter <3.818 cm. Plastic fragments constitute the highest share, with 97.6% of all fragments. The retained plastic ML in the 5 to 50 cm size indicates that the smaller the ML, the higher the ML number (Fig. 5A). This size distribution indicates fragmentation of plastic ML and is a precursor for microplastics[21,47]. Fragmentation is induced by biochemical and physical mechanisms, such as UV light, oxidation, microorganisms, sediment movement, and temperature changes[47]. Fragmentation can occur either during usage with the fragments lost into the environment or within rivers with mechanical fragmentation being the dominant driver due to the river's hydromorphology[47,48]. Based on our results, we cannot evaluate the contribution of either origin to the total amount. In either case, fragments are difficult to remove from rivers because of their small size, and measures are more difficult to implement than for other ML because fragments cannot be assigned to a category, source sector, or use group. Therefore, it is even more critical to reduce ML categories that can be identified as this leads to fewer ML items being exposed to fragmentation processes in rivers. Besides seven categories explicitly focusing on fragments, this study provides benchmarks on annual ML pollution levels for 138 categories of non-fragmented ML in the Rhine to evaluate the effectiveness of reduction measures in the future.

Another option to reduce ML input is to use rising discharge forecasts to direct cleaning efforts. For example, voluntary or communal cleaning activities could be conducted in inundation areas when forecasts predict rising discharges, especially after cultural events such as New Year's Eve. Among the environmental factors, rising discharge is the most relevant factor in increasing ML amount and ML categories in the Rhine (Fig. 4). Previous studies demonstrated that there can be a positive correlation between discharge and ML number, but this correlation is noticeably stronger when accounting for rising discharge[15,25]. This is because rising discharge raises water levels, inundates river banks and shores, and (re-) mobilises ML[15]. Extreme floods, a strong increase in rising discharge, can mobilize up to 100 times more ML than normal water levels[14]. However, our study shows that regular and frequent rising discharge during normal water levels already significantly increases rivers' ML. With floods being a rare event, we assume that floods play a minor role in total ML mobilization compared to normal annual discharge fluctuations. A reason for the strong correlation in our study is the continuous measurement of ML and environmental factors, including 97.7% of the discharge measured in the previous 30 years (Fig. 6). Non-continuous monitoring can lead to data gaps regarding water levels, and visual observation methods can be biased toward larger items[2,25,49]. Regarding the latter, our study shows a stronger correlation of rising discharge with smaller ML than with larger ML (Supplementary Table 2), which could have dampened the correlation coefficients in previous studies. Rising discharge further significantly increased the abundance of plastics, worked wood, metal, and rubber ML. Of these materials, ML that is of relatively small size, low density, and may be used outdoors can be found more often with increasing discharge, for example, plastic caps, plastic lolly and ice cream sticks, rubber condoms, and wooden corks (Supplementary Table 2). In contrast to rising discharge, discharge, rain, and wind, temperature is the only environmental factor negatively correlating with ML categories, especially face masks. This is likely linked to the seasonality found in this study (Fig. 4), which is similar to previous studies[26]. In summer, when temperatures are high and precipitation, wind, and discharge are low, people are more attracted to outdoor activities and might litter closer to the river. In autumn and winter, the increase in rising discharge mobilized ML from the shores and increased ML in the Rhine. This effect could explain two outliers with very high ML and three noticeable peaks of cigarette butts in autumn 2023 during the first larger events of rising discharge after summer. Noticeable is also the difference in the ML category depending on the season. For example, cold temperatures lead to more face masks in the Rhine because masks are primarily worn during the flu season during winter. Overall, rising discharge

may be suitable for predicting ML increases in river systems. Directed cleaning efforts and incentives for proper waste disposal near the river may lead to less ML in the future.

The LT was a robust and reliable physical interception sampling system for floating and submerged ML down to 80 cm for over a year. It processed high flow velocities, adapted to changing discharge volumes, and withstand impacts from large ML and trees. It is, therefore, applicable to other river systems of comparable and larger scales if adequately adjusted. In contrast to the traditional visual counting approach, the LT method allows continuos ML monitoring when frequent removal of retained ML is ensured. More continuous, long-term studies using a litter trap are required to verify the quality gain through this method and revisit previous extrapolation estimates based on different monitoring methods. A current drawback of the LT is its sampling depth of 0.8 m of the water column and the sampling volume of 0.08% of the total discharge volume, and hence a relatively small sample size focused on floating ML. Future LTs could be designed more easily accessible and larger with several depth layers to include the water column and increase the sampling volume. In addition, several LTs can be positioned at multiple locations along the river's length or cross-section to improve spatial resolution and evaluate local ML inputs, for example originating from cultural events or sewer outlets. Continuous, long-term monitoring was crucial for documenting temporal variations in the occurrence of ML. Even though we monitored ML over one year, the same event did not lead to the same ML abundance. i.e., the distinct increase of firework-related ML after New Year's Eve 2022/2023 could not be observed after New Year's Eve in 2023/2024, when the boom was disconnected due to high water levels. Hence, a continuous sampling period is crucial for reliably measuring ML in rivers. Increasing the temporal resolution, e.g., by having shorter cleaning intervals, would enhance the understanding of ML abundance patterns, reduce blockages of the LT, and maintain a more even volume flow through the LT (see SI-1) to prevent the rarely observed cases of ML being flushed out of the front of the LT. Additional filters could extend the monitored ML sizes, e.g., to retain meso- and microlitter, as long as no animals are harmed. The associated resources and labour can be implemented in a citizen science project. The strong commitment by volunteers and funding (about 160.000€) was crucial for maintaining the LT and ML data collection. The successful set-up as a citizen science project allowed us to manage the resource-intensive work while raising awareness, educating volunteers, and engaging the local community through this hands-on experience. Thorough ML monitoring, validation, and safety training ensured high data quality and personal safety. The photo protocol proved beneficial for identifying errors within the validation process. Regarding future monitoring efforts, we recommend using the MSFD list and publishing all raw data and environmental data to accomplish harmonisation. Further extensions of guidelines, for example, by implementing our suggested use, source, and duration groups, could be a step towards more refined monitoring methods and mitigating ML pollution in rivers. In the future, the categories could also be assessed according to their individual risk, and the polymer type could be systematically identified as part of the monitoring procedure. Since the Rhine is located in a similar biome with presumably similar hydro-meteorological characteristics to other rivers[50], such as the Susquehanna River in the USA, the Yellow River in China, the Clutha River in New Zealand, the Biobio River in Chile, the monitoring methods, results, and derived reduction measures can aid in reducing global ML emissions.

## Method
### Location and litter trap design
The Rhine River is Eurasian's ninth largest river[51] with a river basin of 163,645 km²[49] and a population of more than 49 million people[16] in nine adjacent countries[51] (Fig. 1A). With its origin in the Swiss Alps, the Rhine flows northwards into the North Sea with a Delta of three river branches in the Netherlands[51]. According to the Hydroshed dataset[16], 90% of this entire river network and 72% of the population in the river basin live in the upstream area of the LT located in Cologne. The Rhine's discharge, which is influenced by pluvial and nival factors[51], varied from 638 m³ s⁻¹ to 10,700 m³ s⁻¹ in the last 30 years at Cologne gauging station, where the litter trap (LT) is located (1993 to 2022)[52] (Fig. 1A).

On September 8th, 2022, the LT was permanently anchored at Rhine kilometre 690.3 downstream of the Cologne city centre for continuous sampling (50°57′22.2″N 6°58′31,6″E, 50668 Köln, Supplementary Notes 1). The LT was initiated, designed, and operated by the local, non-governmental organisation K.R.A.K.E Kölner Rhein-Aufräum-Kommando-Einheit e.V. The LT functions to remove litter from the Rhine, engage and educate the local community, and monitor macrolitter (ML). The LT is a physical-interception-based method[10] and consists of two stainless steel grid baskets with a floating boom connected to the river bank. Two stainless steel walkways surround the baskets and keep the LT afloat. The total dimensions are 4.7 m in width and 11.19 m in length (Fig. 1B). Each basket measures 2.94 m in width and submerges around 0.8 m depending on flow velocity and waves. The upstream entrance of the first basket, i.e., the cross-sectional sampling area, measures 2.352 m². The cross-sectional sampling area is slightly reduced by vertical rakes (Fig. 1B) that protect the baskets from large, potentially damaging objects, such as trees or timber. The mesh sizes of the steel grids of the baskets vary from 3 × 3 to 5 × 10 cm with the steel being 0.3 cm thick and allow systematic monitoring of objects larger than the diagonal inner diameter of 3.818 cm. Since smaller pieces of litter get caught among the flotsam, the lowest observation size was set at 1 cm. However, due to the mesh size, pieces of <3.818 cm are not fully recorded. The floating boom directs ML into the LT and increases the sampling range of the LT[10]. It was custom-built from floating plastic tubes and submerged steel grid elements (Fig. 1B) with a mesh size of 5 × 20 cm and a depth of 0.6 m. The boom covers a distance of 2 to 9 m to shore, depending on the water level. During a water level higher than 6.05 m, the boom was removed due to the safety concept of the LT.

Design and installation of the LT were assigned by the Federal Waterways and Shipping Administration per regulations related to water shipping and nature conservation. All animals were monitored during the operation to ensure the LT had no adverse effects on the macrofauna. Two alive and unharmed animals (mollusks = 1 (carried to shore), mammals = 1 (*Myocastor coypus*, left on its own), and 68 dead animals (mollusks = 1, amphibians = 1, fishes = 23, birds = 19, mammals = 26) of which ten were decomposed, have been recorded. All dead animals were likely dead before being retained by the LT.

### Sampling and monitoring of macro litter
Biweekly monitoring for this study started on Saturday, September 24th, 2022, and ended on January 13th, 2024 (35 sampling dates). Three additional samplings were required when the LT capacity reached its limit. For maintenance, emptying, and the collection of samples, the LT was approached by boat from the local marina or by kayak from the river bank (Supplementary Fig. 2). Landing nets and fishing waders were used to enter the baskets and collect trapped ML and other objects. Natural objects like branches or leaves were discarded into the river. In contrast, ML was collected in buckets and transported to the monitoring site by boat, car, bicycle trailer, or handcart (Supplementary Fig. 2).

Volunteers and KRAKE members performed ML monitoring as part of the citizen science project, and they were trained and supervised by the scientific team. To ensure high monitoring standards and comparability to other sampling sites and studies, the international MSFD list for European member states was used (also called the J-code list)[17]. Even though the MSFD list was initially developed for marine beaches, its application riverine systems bridges the gap between freshwater and marine ML studies as they are closely linked. The hierarchical organised MSFD list includes nine ML materials (chemicals, cloth/textiles, glass/ceramics, metal, artificial polymer materials (plastics), organic food waste, paper/cardboard, rubber and processed/worked wood), thirteen use groups (agriculture-, aquaculture-, clothing-, building & construction-, food consumption-, fisheries-, hunting-, personal hygiene and care-, medical-, recreational-, smoking-, vehicle-related, and undefined

use), and 183 ML categories (e.g. glass bottles, non-foamed plastic fragments 2.5–50 cm, wooden firework matches)[17]. Measurements that are assigned to each category include ML item count, size class, and wet weight. All information was translated into German and implemented into the ODK Collect app (Get ODK Inc, Version v2022.4.2) to enable fast smartphone monitoring, cloud data storage, and validation by the scientific team.

After each sampling, the ML was sorted according to the categories. The wet weight of small ML was measured for all ML per category with a digital scale (Soehnle, Page Compact 300, max 5 kg, display resolution of 1 g). Large ML was measured with a heavy-duty (Haba Trading B.V., max 300 kg, min 1 kg). Soaked or filled ML, e.g., diapers or bottles, was marked as "weight-impacted." Although ML is defined as objects with a size larger than >2.5 cm[17], the lower size limit was set as 1 cm to include ML, such as cigarette butts that are a category in the MSFD list. Therefore, size was grouped into six classes ((1)/2.5–5 cm, 5–10 cm, 10–20 cm, 20–30 cm, 30–50 cm, and >50 cm) and measured with folding rulers. Metadata includes sampling date, monitoring date, and sampling ID. Finally, a photo was taken for the validation process.

The scientific team validated each dataset entry to reduce observer bias and increase uniformity. Validation and subsequent alterations to the dataset were marked in the raw data according to the validation procedure (Supplementary Methods 1).

Environmental data and social events were measured or obtained from open sources to identify factors mobilizing ML in the Rhine. This includes daily discharge data over the past 30 years (1993 to 2022)[52], daily discharge, and daily water levels for the sampling period from the gauging station in Cologne[53]. Daily wind speed, precipitation, temperature, and sunshine hours were obtained from the Climate Data Center of the German Weather Service[54]. All quantities were resampled to biweekly cumulative sums, except for the temperature used as the biweekly mean. As discharge changes within short periods, the discharge data was decomposed into a falling and rising component for each day. After calculating the change in discharge between the present day and the day before, the cumulative sum of all negative values (falling component) or positive values (rising component) was computed, respectively. Furthermore, the flow velocity was measured with a water flow metre (OTT MF pro, 0-6 m s$^{-1}$, ±2% accuracy) at the entrance, the upstream, and the downstream basket of the LT before and after cleaning at different depths (Supplementary Notes 1). Regarding cultural events, the date was noted for the local festivities, such as the carnival, the Cologne marathon, and New Year's Eve, as these may increase littering. The International Clean-up Day was also recorded as this potentially decreases ML in the Rhine.

## Data analysis, statistics, and extrapolation

Data analysis was performed in the web-based interactive computational environment Jupyter Notebook[55] using Python (Python 3.10.9). The sum of all ML categories determined the total ML number. Since the wet weight could not be gained for each item due to potential artefacts, the estimated total wet weight was derived by the rule of three based on all items without artefacts. The data was analysed for material, size, and category composition. The MSFD list includes 13 usage types, e.g., clothing-related, food-consumption-related, undefined use, smoking-related, and building & construction. However, we grouped the categories into three levels with several subgroups to identify more specific usages, responsibilities, and potential measures. Level one divides the categories into water-based or land-based origin as suggested by previous publications along with the MSFD list[56]. Of this separation, level two delineates four sectors, i.e., private consumer (101 categories), industrial (16 categories), traffic/infrastructure/construction (12 categories), and other items that cannot be linked to a specific source such as fragments (17 categories), similar to Oswald et al. (2023)[57]. Level three further subdivides level two into usage groups such as fragments, drinks-, packaging-, or

smoking-related objects. This classification includes the 13 usage groups provided by the MSFD list and five additional groups, i.e., bags, drink-related, firework-related, fragments, and waste treatment-infrastructure-related. In addition, packaging-related ML (40 categories) and lids (6 categories) were grouped separately as they are subordinate groups of levels two and three.

Furthermore, the ML data was sorted into four classes according to the duration of their consumption phase. Single-use items have a short lifespan, often developed for one-time use and with limited potential for reuse (Directive (EU) 2019/904). Categories assigned as single-use include plastic cotton bud sticks, plastic cutlery, plastic plates and trays, plastic straws, plastic stirrers, plastic food containers made of foamed polystyrene, cups, cup lids of foamed polystyrene, and plastic balloon sticks. The named examples are all banned by the Federal Ministry of Justice in Germany since July 3rd, 2021[58]. In contrast, multi-use items are designed for repeated use, and some may even be refilled or recycled before disposal (Directive (EU) 2018/852). Nevertheless, some ML categories could not be assigned exclusively to single- or multi-use, so they were classified as ambiguous. Categories not assigned to the previous three groups were sorted into non-identifiable.

For statistical analysis, the dataset was subsampled to exactly one year from noon 19.11.2022 to noon 18.11.2023, which is called the statistical year. Within the statistical year, the floating boom was continuously attached to the LT, and no extreme flood events impacted the monitoring. The data of the statistical year was checked for normal distribution and outliers using the Shapiro-Wilk test and graphical analysis, for example with Q-Q plots and histograms (see supplemented code[59]). Outliers were not removed as they all fell within reason of the test setup and data distribution. The dataset was analysed for ML composition patterns including material and group shares, size class distribution, and ML counts (indicated by #) related to seasonality and environmental factors. Spearman rank-correlation was computed due to existing outliers within the dataset ($\rho$ = correlation coefficient, $p < 0.05$). We only included results with strong correlations of $\rho > 0.7$ in the main text. The Mann-Whitney-U test was used to test for significant differences ($p < 0.05$, $r$ = effect size) in ML abundance between seasonal groups.

LT performance, ML discharge for the Rhine's cross-section, and amount of ML transported towards the North Sea were calculated for the ML data of the statistical year by using the mean water level of 283.1 cm and a mean discharge of 1841.3 m$^3$ s$^{-1}$ at Cologne gauging station (see supplementary code[59]). First, ML concentration in the Rhine was calculated by multiplying the submerged LT entrance area (2.352 m²) with the mean flow velocity at the LT entrance (0.66 m s$^{-1}$, Supplementary Notes 1) with the retained ML items number (#) and mass (kg), respectively. The effect of the boom was estimated based on its mesh size that retains only ML >20 cm, its size (on average 5 m long, 0.6 m deep), and the assumption that half of the water that encounters the boom will go below the metal grid. Hence, the share of ML in the LT coming from the boom was determined as 6.8%, which was subtracted from the total ML amount. The corrected ML item and ML mass concentration was the basis for extrapolating total ML at Cologne using three models: (1) ML surface flux, (2) linear ML discharge, and (3) weighted ML discharge. Since all three extrapolation models include continuous data from an entire year, the models describe the spatial extrapolation across the river's width, cross-section, and discharge. Temporal factors, as used in other models[60], were only included in the weighted ML discharge model with regard to flooding events.

ML surface flux was defined as ML items per hour per unit length (ML h$^{-1}$ m$^{-1}$ and ML mass h$^{-1}$ m$^{-1}$). This conversion assumes that all ML in the LT would have also been identified by the visual observation approach used by Gonzalez-Fernandez et al. (2021)[13]. Therefore, ML surface flux per unit length ML was (ML year$^{-1}$ LT width$^{-1}$) multiplied by the Rhine's water-surface width of 326.7 m for total river flux.

The linear ML discharge model extrapolated ML concentration to the mean river discharge at the Cologne gauging station (ML time[-1]). This approach assumed that all ML is distributed equally across the Rhine's cross-section and did not include other influencing factors.

For the weighted ML discharge model, three main factors that may influence the ML distribution in rivers were evaluated: (1) the vertical and horizontal distribution of ML across the river profile, (2) litter deposition and re-suspension, and (3) flooding events.

Since the LT only covered a depth of around 0.8 m and retained floating ML, a vertical distribution factor for the Rhine's profile was introduced. ML with a density lower than water will usually float but may be suspended in the water column due to turbulence in the flow and wind mixing[61,62] or biofilm growth[63]. In return, ML from high-density materials such as metal or glass can float when air is trapped, as was observed multiple times. In the modelling study by Meijer et al. (2021)[11], a global conversion factor accounted for the vertical distribution of the ML in deeper layers of the Rhine. To determine a more Rhine-specific vertical distribution factor, only field studies from river branches in the Delta Rhine were considered. All three rivers, Waal, IJssel and Nieuwe Maas, are part of the European lowlands, heavily modified water bodies and navigable like the Rhine in Cologne[51]. In the Nieuwe Maas, a river branch that connects the Rhine and Meuse rivers with the North Sea, plastics were found twice as often in the deep river layers while the mass concentration was the same[19]. Specifically, the concentration of other litter, such as textiles and rubber, was found more often in deeper layers, which was explained by their higher density[19]. However, this location for data collection is influenced by tide and differs from the LT location in this point. In Dutch branches of the Rhine, Waal, and Ijssel, the vertical distribution factor for plastic ML from surface to deeper water ranged between 0.37 to 3.51, 0.45 to 0.74, and 0.48 to 2.09, respectively[64]. Based on these available studies, a vertical distribution factor of 1.5 accounted for ML that was not floating but suspended in the lower water column or benthic layers. To justify the values applicability, a sensitivity analysis was performed with vertical distribution factors ranging from 1 to 2 with increments of 0.25. Factors influencing ML concentration across the river's left and right sides might have been currents, the bending of the river, and local wind direction[19]. Blondel and Buschmann (2022)[19] found that the plastic litter mass was 73% higher on the leeward side of the wind than on the windward, with a maximum of 300%. Vriend et al. (2020)[18] did not find differences in plastic concentration of the Rhine's left and right side related to wind during visual observation of ML. Spiral flows in a river's bend typically point toward the outer bend of the river, which would increase the ML concentration on that side. This would explain the behaviours of ML mimicking drifters that were observed to travel toward the outside half of a river bend[65]. The LT was positioned on the outside half on a slight right bend on the leeward, as winds typically come from the west. Based on these two factors, which were difficult to quantify and presumably cancel each other out, a weighting factor for a horizontal distribution was not used for the weighted ML discharge model. Deposition and resuspension are relevant factors that influence short-term concentrations of macroplastics[22] and microplastics[62] in rivers. Still, no quantitative and long-term data is available for ML deposition and resuspension in rivers. Field studies on ML trajectories and transportation rates in rivers showed that drifters were retained 23 times, spending an average of 80% of their time stuck in ports and other obstacles during the on average 23-day testing period[65]. Stranding periods were observed to last up to ten days[66]. Both studies indicate deposition and resuspension behaviour that prolong the transportation time in rivers but do not support the assumption of permanent removal from the river, which is likely to happen only through clean-up activities[65]. Additionally, waste generation in Germany was relatively constant in the last 20 years, ranging between 406.7 Mt and 356.1 Mt[67]. Therefore, a continuous ML input into the Rhine and a steady state of ML deposition-resuspension were assumed for the extrapolation of statistical year data. Extreme floods are rare events that may also impact mobilisation, deposition, and resuspension in rivers[14], for example, by flooding litter bins (Fig. 6A). During the extreme floods in Western Germany in July 2021, the Rhine discharge at Cologne gauging station was 6490 m³ s[-1], which equals an exceedance probability of 0.41% according to a 30-year flow duration curve based on the medium daily discharge at Cologne gauging station[52] (Fig. 6B). An exceedance probability of 0.5% means that the corresponding discharge was statistically occurring once every 200 days. During the same event, the Dutch Meuse showed an increase in ML flux by a factor of 141 to 363 compared to non-flooding conditions[14]. In the statistical year, the highest discharge was 5010 m³ s[-1] with an exceedance probability of 2.32%. Therefore, the statistical year represents the most common discharge levels without extreme floods and does not cover unusual mobilisation of ML. Assuming that flooding as well as strong rainfall events were covered by changes of rising discharge, we introduced a rising discharge factor to account for this gap in the extrapolation. The rising discharge factor was based on the difference of the mean annual rising discharge of the previous 30-years with 21,595 m³ s[-1], thus, also covering the extreme flood events from 1993 and 1995 in Cologne, and the mean rising discharge of the statistical year. This resulted in a rising discharge factor of 1.0424 for the weighted model. Note that a potential non-linear increase of ML during extreme flooding was not considered as this data was still missing and needs to be examined in future studies.

As riverine ML may end up in the ocean[26], we calculated the ML transport towards the North Sea. Therefore, ML items and mass of the linear and weighted ML discharge estimates were calculated per person, since waste generation is linked to anthropogenic influence[26,27], and then multiplied by the total Rhine-associated population.

Based on fragmentation processes it was assumed that the number of plastic litter increases with decreasing size[47]. To verify this assumption for the ML in the Rhine, the number and mass concentration of plastic ML were plotted against the size classes and fitted with a quadratic model.

The underlying code for this study is available at Zenodo and can be accessed via this link https://doi.org/10.5281/zenodo.17108281.

### Reporting summary

Further information on research design is available in the Nature Research Reporting Summary linked to this article.

### Data availability

The raw data is available at doi: 10.5281/zenodo.17108281 (link: https://doi.org/10.5281/zenodo.17108281).

### Code availability

The code is available at doi: 10.5281/zenodo.17108281 (link: https://doi.org/10.5281/zenodo.17108281).

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

## Acknowledgements

We highly appreciate the help of all KRAKE e.V. members, volunteers, supporters and sponsors. We want to especially thank Carsten Florich, Frank Fuchs, Fabian Heinisch, Kai Hirsch, Thorsten Kniewel, Jana Langensiepen, Martina Lorenz, Katrin Maschke, Judith Müller, Sabrina Scholtisseck, Christian Stock for their commitment and the support from Deutz Marina for being able to disembark in Cologne Deutz. We thank the workgroups of Prof. Ternes, Prof. Evers, Prof. Hochschild and Prof. Blanke for their support and advice during the study and manuscript writing. We also thank Dr. Clarissa Glaser for hydrological advice, Dr. Christian Sommer for statistical advice, and Daniela Bornstein and Prof. Johannes Steinhaus for advice in material science. None of the authors received direct funding from the non-profit K.R.A.K.E e.V. which is an organization funded primarily by membership fees, donations and sponsors. K.R.A.K.E. e.V. did provide resources such as the litter trap, scales, container, ship engine, fuel and phones. The boat used to transport people and litter from the shore to the litter trap was made available to KRAKE e.V. by the Cologne Rheinau Marina. LH was supported by the European Research Council (ERC) under the European Union's Horizon 2020 research and innovation program (grant agreement no. 54290) and Horizon 2020 proof of concept program (grant agreement no. 101123148). The funders played no role in study design, data collection, analysis and interpretation of data, or the writing of this manuscript.

## Author contributions

Nina Gnann and Katharina Höreth contributed equally to this manuscript. Nina Gnann was involved in conceptualization, data curation, formal analysis, investigation, methodology, software, validation, visualisation, and writing - original draft. Katharina Höreth was involved in conceptualization, data curation, formal analysis, investigation, methodology, software, validation, visualisation, and writing - original draft. Nicolas Schweigert contributed to conceptualization, funding acquisition, investigation, methodology, project administration, resources, supervision, and writing - review & editing. Mariele Evers contributed to supervision, funding, and writing - review & editing. Thomas A. Ternes to supervision, and writing - review & editing. Leandra Hamann was involved in conceptualization, formal analysis, investigation, methodology, supervision, visualisation, and writing - original draft. All authors read and approved the final manuscript.

## Funding

## Competing interests

K.H. and N.S. hold unpaid memberships in the non-governmental organisation K.R.A.K.E. e.V. We acknowledge that N.S. is affiliated with Bayer AG during the conduct of this study and the preparation of the manuscript. However, no funding was received from Bayer AG, and there is no conflict of interest to declare. We also acknowledge that N.G. is affiliated with the Zweckverband Bodensee-Wasserversorgung during the conduct of this study and the preparation of the manuscript. However, no funding was received from the Zweckverband Bodensee-Wasserversorgung, and there is no conflict of interest to declare.
