## [Transparent Peer Review file · Communications Sustainability]

The river Rhine transports around 4,000 tons of macrolitter towards the North Sea each year

Corresponding Author: Dr Leandra Hamann

Version 0:

Decision Letter:

Dear Dr Hamann,

Your manuscript titled "Underestimated Pathways: Long-Term Monitoring Reveals Quantities, Composition, Sources and Environmental Drivers of Riverine Macrolitter" has now been seen by 3 reviewers, and we include their comments at the end of this message. They find your work of interest, but some important points are raised. We are interested in the possibility of publishing your study in Communications Sustainability, but would like to consider your responses to these concerns and assess a revised manuscript before we make a final decision on publication.

We therefore invite you to revise and resubmit your manuscript, along with a point-by-point response that takes into account the points raised. Please highlight all changes in the manuscript text file.

In addition, kindly discuss the methodological uncertainty transparently.

Please submit your point-by-point responses as a separate file, distinct from your cover letter where you can add responses to the Editors' comments that you do not want to be made available to the reviewers. Word files are preferred. We recommend that any figures, tables or graphs that are included in the response to reviewers are also included in the main article or Supplementary Information.

Please use the following link to submit your revised manuscript, point-by-point response to the referees' comments (which should be in a separate document to any cover letter), a tracked-changes version of the manuscript (as a PDF file) and the completed checklist:

Link Redacted

We hope to receive your revised paper within six weeks; please let us know if you aren't able to submit it within this time so that we can discuss how best to proceed. If we don't hear from you, and the revision process takes significantly longer, we may close your file. In this event, we will still be happy to reconsider your paper at a later date, as long as nothing similar has been accepted for publication at Communications Sustainability or published elsewhere in the meantime.

Please do not hesitate to contact us if you have any questions or would like to discuss these revisions further. We look forward to seeing the revised manuscript and thank you for the opportunity to review your work.

Best regards,

Nandita Basu, PhD

EDITORIAL POLICIES AND FORMATTING

- Behavioural and social science
- Ecological, evolutionary & environmental sciences
- Life sciences

Furthermore, please align your manuscript with our format requirements, which are summarized on the following checklist: <https://www.nature.com/documents/commsj-phys-style-formatting-checklist-article.pdf> Communications Sustainability formatting checklist

and also in our style and formatting guide <https://www.nature.com/documents/commsj-phys-style-formatting-guide-accept.pdf> Communications Sustainability formatting guide .

*** DATA: Communications Sustainability endorses the principles of the Enabling FAIR data project (<http://www.copdess.org/enabling-fair-data-project/>). We ask authors to make the data that support their conclusions available in permanent, publicly accessible data repositories. (Please contact the editor if you are unable to make your data available).

All Communications Sustainability manuscripts must include a section titled "Data Availability" at the end of the Methods section or main text (if no Methods). More information on this policy, is available at <http://www.nature.com/authors/policies/data/data-availability-statements-data-citations.pdf>.

If a community resource is unavailable, data can be submitted to generalist repositories such as <https://figshare.com/> or <http://datadryad.org/> Dryad Digital Repository. Please provide a unique identifier for the data (for example a DOI or a permanent URL) in the data availability statement, if possible. If the repository does not provide identifiers, we encourage authors to supply the search terms that will return the data. For data that have been obtained from publicly available sources, please provide a URL and the specific data product name in the data availability statement. Data with a DOI should be further cited in the methods reference section.

<http://www.nature.com/authors/policies/availability.html>.

REVIEWER COMMENTS:

Reviewer #1 (Remarks to the Author):

This study presents a comprehensive and long-term monitoring of riverine macrolitter (ML) in the Rhine River using a custom-designed litter trap, coupled with citizen science participation. The research addresses critical knowledge gaps in ML dynamics, particularly regarding non-plastic materials, source attribution, and environmental drivers. The findings have significant implications for policy and management, emphasizing the need for holistic approaches beyond plastic-focused strategies. The following suggestions can be considered:

1. The litter trap's mesh size (4.24 cm diagonal) may under-sample small ML (<5 cm, 23% of observed items), potentially

skewing fragment representation. Clarify how this limitation affects material composition estimates, especially for plastics (e.g., microplastic precursors).

2. The study relies on a single fixed monitoring site in Cologne, which, while covering 72% of the basin's population upstream, may not fully represent the entire Rhine's macro-litter (ML) distribution. For example, ML sources differ significantly between mountainous upstream regions (e.g., agricultural plastic mulch) and downstream delta areas (e.g., industrial rubber/metal waste). The lack of spatial differentiation (e.g., urban vs. Rural, tributary contributions) limits the generalizability of source attribution and transport patterns.

3. The model assumes uniform ML distribution across the river's cross-section, ignoring effects of channel geometry (e.g., helical flow, bend-induced sedimentation) that concentrate ML on outer river banks (as noted in Line 667). This oversight may skew estimates of ML transport, particularly in meandering reaches or areas with variable flow velocities.

2. While training and validation are mentioned, provide details on inter-observer variability (e.g., kappa statistics for category assignments) to strengthen methodological rigor.

3. The weighted ML discharge model uses a vertical distribution factor (1.5) derived from adjacent rivers (Nieuwe Maas). Justify its applicability to the Rhine's hydromorphology or include sensitivity analyses for this parameter.

4. Figure 2 (ML composition) and Table 3 (size classes) are complex; consider splitting into sub-figures or using standardized color codes for material categories to improve readability.

5. In Figure 3 (temporal trends), overlay key cultural events with ML spikes to explicitly demonstrate event-driven littering.

6. Clarify the rationale for excluding extreme outliers in correlation analyses (Line 178) and assess their impact on results (e.g., sensitivity tests).

7. Discuss limitations of the litter trap design (e.g., sampling depth, mesh size) and how future studies might address them (e.g., multi-depth traps, as suggested in Line 397).

8. Expand the comparison with global rivers (e.g., Yangtze, Danube) using standardized metrics (e.g., ML items/person/year) to contextualize the Rhine's role in marine pollution (Line 250–251).

9. Integrate recent literature on ML fragmentation (e.g., Liro et al., 2023) to discuss how size-class distribution (Figure 2C) reflects in-river degradation processes.

10. Propose specific interventions for top source sectors (e.g., private consumers) such as deposit-return systems for glass bottles (5.3% of ML) or regulatory targets for firework waste (10.7% of ML), as mentioned in Line 304.

Classification System and Data Standardization Gaps

11. 54% of ML is classified as "undefined use" (Figure 2E), reflecting limitations in categorizing fragments or ambiguous items. This introduces bias in source attribution (e.g., conflating industrial vs. consumer-derived fragments). Incorporating technologies like FTIR spectroscopy for material analysis could improve classification accuracy.

Compatibility with International Standards: While adopting the MSFD list, the novel "source sector" and "durability" groupings are not yet widely standardized, potentially limiting comparability with global studies. The paper should clarify how these additions align with or enhance existing frameworks (e.g., UN Marine Litter Guidelines).

12. Highlight the EU's upcoming tethered cap regulation (Line 356) and estimate its potential impact using study data (e.g., reduction in plastic caps/lids, 4.5% of ML).

13. The study region (Germany) has robust recycling systems (e.g., deposit-return for glass/plastic), but the paper does not assess how policy effectiveness (or gaps like illegal dumping, overflowing bins during floods, Figure 6A) influences ML composition (e.g., why glass bottles still account for 5.3% of ML despite regulations).

14. The statistical year excluded extreme floods (e.g., 2021 Rhine flood), yet the discussion notes floods can mobilize "100× more ML than normal levels" (Line 652). The model uses a modest "rising discharge factor" (1.0424) to account for this, but without quantifying rare high-flow events' contribution to annual transport (e.g., sensitivity analysis with historical flood data is missing).

Climate Change Omissions: Future increases in extreme precipitation in the Rhine basin may alter seasonal ML transport (e.g., more winter peaks), but the study does not engage with climate model projections or discuss long-term trend implications.

Reviewer #2 (Remarks to the Author):

The paper provides data from a continuous 16-month monitoring of macro-litter using a litter trap in the Rhine River. The monitoring approach including participation of citizen scientist is explained in detail and appropriate for the task. The data assessment and statistics are well done, plausible and justified. A thorough discussion highlights the importance of continuous monitoring and compares the findings with those from other studies. The new data base for the amount of macro-litter transported by the Rhine River and disposed to the Northern Sea, the experiences gained using the litter trap and the well prepared discussion are a valuable contribution to science and to conclude on measures to minimize macro-litter emissions.

The structure of the manuscript has to be corrected, chapter 2 (Results) has got a wrong number and should be replaced by chapter "Methods" (line 433ff). Chapter 4 should be Discussion.

Some minor comments: L48 "a rise in human carcinogenic toxicity" would need a more specific reference additionally to the related general reference [7].

Fig. F: legend central pie chart: infrastructure .. construction

L140: Please be precise and make clear that not sampling dates are compared but sampling periods of two weeks.

L216: 4 Discussion

L270: .. zinc, other heavy metals ..

L308: A comment on potential risks from wooden corks may be added. As it is named together with other litter known to results in risks for nature etc., the potential impact of wooden corks is less obvious.

L502: Not clear what is d = 1 g, Do you mean min 1 g?

L693ff: Adjust format of reference according to journal guidelines, check ref. [10]-not listing all authors; [16]-add web-link; [18,

51]-vol/no of journal missing; The Science of The Total Environment
Supplementary Information:
L28: The litter trap (LT) ..
L42, 43: add blank between numbers and units
L51 vs. L 54: 6 sampling dates or N=5?
L69: Fig. S12.1
L72: (C: or (c) for copyright?

Reviewer #3 (Remarks to the Author):

The manuscript presents a thorough and innovative study on riverine macrolitter (ML) in the Rhine River, one of Europe's largest and most significant waterways. Over a 16-month period, the authors implemented a continuous monitoring strategy using a custom-designed physical-interception litter trap, which was maintained and operated in collaboration with local citizen scientists. This approach enabled the collection of 20,339 floating ML items larger than 1 cm, which were then classified using internationally recognized standards.

The dataset reveals that private consumers are the predominant source of ML, accounting for 56.4% of the total, with plastics comprising 69.7% of items by count but only 14.8% by weight. This highlights the significant role of non-plastic materials in the overall mass of ML. The authors also demonstrate that river discharge is a strong predictor of ML abundance, suggesting that hydrological conditions play a crucial role in the mobilization and transport of litter. Based on their data, the authors estimate that the Rhine transports between 27 and 42.2 million ML items (3,010.5 to 4,707.5 tons) annually toward the North Sea—figures that substantially exceed previous European estimates. The study documents high temporal variability in ML abundance, with biweekly sample counts fluctuating by a factor of 41, underscoring the importance of long-term and continuous monitoring to accurately capture the dynamics of riverine litter.

One of the manuscript's greatest strengths lies in its methodological rigor and innovation. The use of a custom-built litter trap capable of intercepting both floating and submerged ML up to 80 cm below the surface represents a significant advance over traditional visual observation methods, which are often limited by field conditions and observer bias. The long-term, biweekly sampling regime provides a robust dataset that captures both seasonal and event-driven variability. The integration of citizen science not only made extended monitoring feasible but also fostered local engagement and awareness. The comprehensive classification of ML by material, use, source, and sector—following the MSFD list and further refined by the authors—enhances the relevance of the findings for both scientific and policy audiences. The statistical analysis is sound, with clear evidence presented for the influence of environmental drivers such as discharge, precipitation, and wind speed on ML abundance and composition.

Despite these strengths, the study has some limitations. The mesh size of the litter trap precluded the systematic collection of ML smaller than 5 cm, which may lead to an underestimation of the contribution of smaller macrolitter and microplastics. Additionally, a significant portion of the collected ML (about 30.6%) could not be definitively assigned to a source or use category, primarily due to the prevalence of fragments and ambiguous items, limiting the precision of source attribution. The annual transport estimates are based on a single year that did not include extreme flooding events, so they may not fully capture interannual or episodic variability that could significantly influence ML fluxes. The study's spatial scope is limited to a single site in Cologne, which, while strategically chosen, may not reflect spatial heterogeneity along the river's course. Furthermore, aside from a clear spike in firework-related litter following New Year's Eve, the study was unable to link other cultural events to changes in ML abundance, possibly due to the overriding influence of hydrological factors or insufficient event-specific data.

The manuscript is clearly written and well-organized, with informative figures and tables that effectively support the narrative. The use of international classification standards and the detailed breakdown of ML types add clarity and facilitate comparisons with other studies. The findings have important policy implications—particularly the strong correlation between ML abundance and river discharge—which could inform targeted clean-up efforts and monitoring strategies. The study underscores the need for harmonized, long-term monitoring of riverine ML to support EU and international marine litter reduction goals.

To further strengthen the work, future studies could complement the current methodology with approaches targeting smaller ML and microplastics, expand sampling to additional sites along the river to assess spatial variability, and incorporate longer-term and event-based monitoring to refine annual flux estimates and better capture the impact of extreme events. Advanced forensic or chemical analysis methods could also be explored to improve the identification of ambiguous or fragmentary ML items.

Specific comments

1. The manuscript notes that the litter trap's mesh size prevented systematic sampling of ML items smaller than 5 cm, which represent 23% of the collected items but were not fully accounted for. Please provide a more detailed discussion on how this limitation may have influenced the representativeness of your results, particularly regarding the total ML flux and the composition of smaller litter classes. Consider suggesting how future studies might address this gap.
2. A significant proportion of the collected ML (about 30.6%) could not be assigned to a specific source or use group, mainly

due to fragments and ambiguous items. Please elaborate on the challenges faced in source attribution and discuss whether additional analytical techniques (e.g., chemical fingerprinting, forensic analysis) could help improve the identification of these ambiguous items in future research.

3. The annual ML transport estimates are based on a single year without extreme flooding events. Please discuss the potential variability in ML fluxes that might occur during years with atypical hydrological conditions, such as major floods or droughts. Clarify how representative your sampling year is for long-term trends and consider including a sensitivity analysis or uncertainty estimate.

4. The study is based on a single sampling site in Cologne. Please discuss the extent to which this location is representative of the entire Rhine River and whether local factors (e.g., urban density, tributary inputs, river engineering) may have influenced your findings. Suggest how additional sites along the river could enhance the spatial resolution and generalizability of the results.

5. While a clear spike in firework-related litter was observed after New Year's Eve, the manuscript states that other cultural events could not be linked to changes in ML abundance. Please clarify the methods used to identify and analyze the impact of cultural events. Consider whether more granular event-specific data or targeted sampling during known high-activity periods could provide further insights into the influence of human activities on riverine ML.

** Visit Nature Portfolio's author and referees' website at www.nature.com/authors for information about policies, services and author benefits**

Communications Sustainability is committed to improving transparency in authorship. As part of our efforts in this direction, we are now requesting that all authors identified as 'corresponding author' create and link their Open Researcher and Contributor Identifier (ORCID) with their account on the Manuscript Tracking System prior to acceptance. ORCID helps the scientific community achieve unambiguous attribution of all scholarly contributions. You can create and link your ORCID from the home page of the Manuscript Tracking System by clicking on 'Modify my Springer Nature account' and following the instructions in the link below. Please also inform all co-authors that they can add their ORCID to their accounts and that they must do so prior to acceptance.

Version 1:

Decision Letter:

<*** REMEMBER TO ATTACH REVISIONS CHECKLIST (WORD)***>

Dear Dr Hamann,

Your manuscript titled "Underestimated Pathways: Long-Term Monitoring Reveals Quantities, Composition, Sources and Environmental Drivers of Riverine Macrolitter" has now been seen by our reviewers, whose comments appear below. In light of their advice we are delighted to say that we are happy, in principle, to publish a suitably revised version in Communications Sustainability.

We therefore invite you to revise your paper one last time to address the remaining concerns of our reviewers. At the same time we ask that you edit your manuscript to comply with our format requirements and to maximise the accessibility and therefore the impact of your work.

EDITORIAL REQUESTS:

*****Please take care to match our formatting and policy requirements. We will check revised manuscript and return manuscripts that do not comply. Such requests will lead to delays. *****

SUBMISSION INFORMATION:

OPEN ACCESS:

Communications Sustainability is a fully open access journal. Articles are made freely accessible on publication. For further information about article processing charges, open access funding, and advice and support from Nature Portfolio, please visit <https://www.nature.com/commssustain/open-access>

Link Redacted

Best regards,

Nandita Basu, PhD
Associate Editor, Communications Sustainability
Consulting Editor, Communications Earth & Environment
Nature Portfolio

REVIEWERS' COMMENTS:

Reviewer #1 (Remarks to the Author):

The author has solved all the problems and I have no new issues. I suggest publishing this paper.

Reviewer #2 (Remarks to the Author):

All comments have been addressed to full satisfaction, including those of the other two reviewers. The manuscript has been further improved and is now ready for publication.
Thank you for your great work - sampling is always challenging and time consuming but the backbone of all modeling studies.

Reviewer #3 (Remarks to the Author):

Thank you for sharing the revised manuscript along with the detailed point-by-point response to reviewer comments. Overall, I commend the authors for addressing all comments thoroughly and providing a well-reasoned and convincing rebuttal. All concerns have been satisfactorily resolved, and I am pleased to recommend acceptance of the manuscript in its current form.

** Visit Nature Portfolio's author and reviewers' website at <http://www.nature.com/authors> for information about policies, services and author benefits**

REVIEWER COMMENTS:

Reviewer #1 (Remarks to the Author):

This study presents a comprehensive and long-term monitoring of riverine macrolitter (ML) in the Rhine River using a custom-designed litter trap, coupled with citizen science participation. The research addresses critical knowledge gaps in ML dynamics, particularly regarding non-plastic materials, source attribution, and environmental drivers. The findings have significant implications for policy and management, emphasizing the need for holistic approaches beyond plastic-focused strategies. The following suggestions can be considered:

1. The litter trap's mesh size (4.24 cm diagonal) may under-sample small ML (<5 cm, 23% of observed items), potentially skewing fragment representation. Clarify how this limitation affects material composition estimates, especially for plastics (e.g., microplastic precursors).

Dear Reviewer, thank you very much for these detailed questions and comments on our manuscript that help us to improve its quality.

We agree that an underestimation due to mesh size is a very important point. This limitation will mostly effect ML number but less ML weight, so the data points in Figure 5A at 50 mm would probably deviate even more with a smaller mesh size. We have now added an asterisk in the figure 5A and expanded the figure caption to highlight that point. It now says in line 220ff: "The asterisk (*) indicates that this size class is smaller than the systematic sampling size and the calculated concentration for <50 mm is likely underestimated."

Figure 2 is based on ML numbers and is potentially also affected by a skewed representation of small ML. Therefore, we added text on line 289f: "With a share of 69.7% by number, which presumably underestimates the share of small ML due to the mesh size of 4.24 cm, we also found plastics to be the most common material in the Rhine" and added "by number" to the Figure 2 caption in line 132.

Regarding the influence on the models, we have added a few sentences in line 257ff: "Regarding the size distribution, the total estimates are probably underestimating ML under 5 cm as 4.24 cm is the lower limit of the systematic sampling size. Small ML below this size limit, such as fragments, cigarette butts, bottle caps, lids or toys, have a share of 23% of all ML by number (Figure 2C). Based on our finding that smaller ML is higher in number and lower in weight (Figure 5A), this underestimate of ML will more effect the extrapolation of total ML number than total ML weight transported in the Rhine."

The problem that this especially influences the share of fragments is described in line 375ff. We have now added a sentence regarding microplastics in line 376f saying: "This size distribution indicates fragmentation of plastic ML and is a precursor for microplastics".

To solve this problem in the future, we wrote that it would be beneficial to add smaller filters to the LT to retain meso- and microlitter in line 456ff.

2. The study relies on a single fixed monitoring site in Cologne, which, while covering 72% of the basin's population upstream, may not fully represent the entire Rhine's macro-litter (ML) distribution. For example, ML sources differ significantly between mountainous upstream regions (e.g., agricultural plastic mulch) and downstream delta areas (e.g., industrial

rubber/metal waste). The lack of spatial differentiation (e.g., urban vs. Rural, tributary contributions) limits the generalizability of source attribution and transport patterns.

Thank you for bringing up this point. We agree that a single fixed monitoring site does not allow for any source attribution across the river basin. Therefore, spatial distribution was not within our scope of this publication, as mentioned in line 444ff. A second litter trap will be implemented, probably next year in Wiesbaden and other studies investigating ML along the river shores are planned in the future to gain more data for the spatial differentiation.

3. The model assumes uniform ML distribution across the river's cross-section, ignoring effects of channel geometry (e.g., helical flow, bend-induced sedimentation) that concentrate ML on outer river banks (as noted in line 667).

Yes, we agree with your statement that the weighted model is not considering factors like helical flow, shipping and other factors. Unfortunately, there is only limited data available to make good assumptions. We discuss the influence of currents, river bends and wind in line 673ff to reason why we did not add such factor to the model. Additionally, we discuss deposition and re-suspension of ML on the river shores in line 683ff.

We also added two sentences in the Discussion in line 249ff to emphasize this problem and identify this data gap: "More data needs also to be gained for the horizontal and vertical distribution of ML in rivers, that are influenced by ML density, wind, river bends, currents, or shipping. The sensitivity analysis of the vertical distribution factors shows that the extrapolation models is quite sensitive to this factor (Figure 5B), which leads to potential under- or overestimates of ML along the vertical dimension."

2. While training and validation are mentioned, provide details on inter-observer variability (e.g., kappa statistics for category assignments) to strengthen methodological rigor.

Thank you for making this point. Every item was validated by either NG and/or KH based on photos. The items were compared with the online photo catalogue of the Joint List of Litter Categories of the EU and the description in the list itself (see <https://mcc.jrc.ec.europa.eu/main/photocatalogue.py?N=41&O=457&cat=all>). If any doubts remained, NG, KH and LH discussed the case together and made a joint decision on how to categorize the data. We have now described this procedure more clearly in the SI-3 line 92f: "Difficult cases were discussed and validated by three authors (NG, KH, and LH)". Additionally, we are publishing the photos of the litter along with the raw data and changes made through the validation process, so the decisions made by us are completely transparent to the reader. Because of this strategic procedure and involvement of several observers, we did not calculate inter-observer variability.

3. The weighted ML discharge model uses a vertical distribution factor (1.5) derived from adjacent rivers (Nieuwe Maas). Justify its applicability to the Rhine's hydromorphology or include sensitivity analyses for this parameter.

Thank you for suggesting these improvements. We included a sensitivity analysis for the vertical distribution factor ranging from 1 to 2 with increments of 0.25. This range was derived from the available data from previous studies as described in line 666ff.

We added information for different vertical distribution factors on the weighted extrapolation model in Figure 5B to make the variations visible, added “The weighted extrapolation model was calculated with a vertical distribution factor ranging from 1 to 2 in increments of 0.25 (green bars) around the base value of 1.5 (dark blue bar).” to the Figure 5B caption, and discuss the quality of the models in line 250ff in the discussion: “The sensitivity analysis of the vertical distribution factors shows that the extrapolation model is quite sensitive to this factor (Figure 5B), which leads to potential under- or overestimates of ML along the vertical dimension. For example, a change of vertical distribution factor from 1 to 2 duplicates the ML discharge. In another study on the estuaries of three other rivers in Germany, Ems, Weser, and Elbe, the number of ML items increased by a factor of 1.3 up to 292.4 from the surface to the water column depending on the river and ambient conditions²⁶, which shows a high variance”.

For explaining the base vertical distribution factor of 1.5, we selected studies exclusively from rivers which are branches of the Rhine river itself, downstream of the LT location, namely the river Waal, IJssel and Nieuwe Maas. The river Waal and IJssel are two of the three branches the Rhine divides into when reaching the Delta Rhine in the Netherlands. According to Wantzen et al. (2022) the Waal covers 65% and the IJssel 12% of the discharge of the Rhine’s main stream. The Nieuwe Maas is located further downstream in the Rhine delta and flows through the city of Rotterdam into the North sea. More than 80% of the Nieuwe Maas’s discharge is determined by the Rhine (Bondel & Buschmann 2022). All rivers are also highly modified and navigable. Both the Lower Rhine, where the LT is located, and the Delta Rhines have a gentle slope as part of the European Lowlands (Wantzen et al. 2022). According to Wantzen et al. (2022) the average annual discharge upstream of the Lower Rhine in Germany is 2043 m³/s (Wantzen et al. 2022). In comparison the average annual discharge upstream of the Delta Rhine is 2252 m³/s (Wantzen et al. 2022). Based on this comparison, we assume that type of ML and hydromorphology, which both influence ML vertical distribution, are similar within the same river system. In the main text, we describe that in line 654ff. It now says: “To determine a more Rhine-specific vertical distribution factor, only field studies from river branches in the Delta Rhine were considered (Waal, IJssel and Nieuwe Maas). All three rivers, Waal, IJssel and Nieuwe Maas, are part of the European lowlands, heavily modified water bodies and navigable like the Rhine in Cologne (Wantzen et al. 2022).”

4. Figure 2 (ML composition) and Table 3 (size classes) are complex; consider splitting into sub-figures or using standardized color codes for material categories to improve readability.

Thank you for the suggestions. We have changed the colours in Figure 2D and matched them with the same categories in Figure 3B. These categories should be now more distinctive from the other graphs in the same figure. We decided against splitting up the figures as they are grouped together by themes. We are not sure what Table 3 you are referring to.

5. In Figure 3 (temporal trends), overlay key cultural events with ML spikes to explicitly demonstrate event-driven littering.

We have added the cultural events to figure 3A so they can be better related to the discharge and ML peaks shown in that graph.

6. Clarify the rationale for excluding extreme outliers in correlation analyses (Line 178) and assess their impact on results (e.g., sensitivity tests).

Thank you for this comment. As described in line 609ff, we have checked the data for outliers but not removed any as all data points were within reason of the test setup. We adjusted the text in line 612f: „Outliers were not removed as they all fell within reason of the test setup and data distribution.“

Regarding the correlation analyses of ML number related to environmental parameters, e.g., discharge, we decided to use the Spearman test instead of the Pearson test as it is more robust to outliers within the data. For example, the numbers of the category of worked wood was relatively constant across the statistical year, except one >1000 increase after New Year's Eve (see the following inserted graphs). Here, the Spearman test clearly shows the better statistical fit with $\rho=0.89$ and $p=1.885e-09$. (Pearson: $\rho=0.23$, $p=0.26$). Plastic ML numbers (see the second inserted graph), which exhibit a more uniform distribution, have a Spearman $\rho=0.78$ and a Person $\rho=0.79$, with both significance levels (p) well below 0.05.

7. Discuss limitations of the litter trap design (e.g., sampling depth, mesh size) and how future studies might address them (e.g., multi-depth traps, as suggested in Line 397).

We have tried to elaborate the limitations of the LT a bit more and added aspects such as accessibility in addition to the previously mentioned factors of low sampling depth, low sampling volume and mesh size, and improvement through multi-depth layers, more locations and more frequent cleaning, i.e., sampling, and additional filters in line 443ff.

8. Expand the comparison with global rivers (e.g., Yangtze, Danube) using standardized metrics (e.g., ML items/person/year) to contextualize the Rhine's role in marine pollution (Line 250–251).

So far, we have drawn comparisons to other rivers based on items per hour to the Danube, Vistula and Adour (line 270-272), and to the Yangtze River (272-275) based on tonnes per year and kg/year/person. Based on these statements we conclude that the Rhine contributes more to ML pollution than previously estimated. We are hesitant to expand this comparison because the sampling and monitoring methods are different -and have also led to underestimated ML transport in previous studies - to the continuously sampling approach we introduce here. To explain our position, we have added the following text in line 279ff: "However, these comparisons are limited by the fact that previous estimates of total ML transport in other rivers were based on non-continuous sampling methods. These need to be reviewed and potentially redone in the future to allow comparison on harmonised sampling methods and determine a more precise contribution of each river to global ML transport into the sea."

9. Integrate recent literature on ML fragmentation (e.g., Liro et al., 2023) to discuss how size-class distribution (Figure 2C) reflects in-river degradation processes.

We agree that Liro et al., 2023 is a valuable source and it was already included in the text as ref. number 51 (in revised manuscript), for example in line 376f; 725f. In line 376-382, the reference is specifically mentioned in the context of fragmentation and fragmentation processes due to the river's hydromorphology.

10. Propose specific interventions for top source sectors (e.g., private consumers) such as deposit-return systems for glass bottles (5.3% of ML) or regulatory targets for firework waste (10.7% of ML), as mentioned in Line 304.

Thank you for making this point. We have expanded our text about glass bottles and the German deposit system with following suggestion in line 347ff: "One option to further decrease glass bottles in the Rhine could be to extend the deposit system to spirits, wine and sparkling wine bottles, which currently are not subject to a deposit."

We have also moved the text about bottle caps to line 359ff and expanded it to: "Newly introduced regulations offer a great opportunity to evaluate their effectiveness. This includes the EU regulation on plastic caps/lids in which all lids of single-use drinking containers up to 3 litres must be fitted with a tethered cap from July 2024 onwards (Directive (EU) 2023/1060). Based on our data, the regulation can potentially reduce 4.5% of floating ML in the Rhine. If all six lid categories were considered, including wooden corks and metal lids, the EU regulation could reduce ML by 7.3 % (1473 #). In case this regulation leads to a significant reduction of ML items, further regulations or bans could be supported by these findings, for example for fireworks related ML".

11. Classification System and Data Standardization Gaps

54% of ML is classified as "undefined use" (Figure 2E), reflecting limitations in categorizing fragments or ambiguous items. This introduces bias in source attribution (e.g., conflating industrial vs. consumer-derived fragments). Incorporating technologies like FTIR spectroscopy for material analysis could improve classification accuracy.

Yes, we agree the 54% of undefined use is very high. Figure 2E picturing this high number is based on the given classification system of the MSFD developed by Fleet et al. (2021). We wanted to show that in comparison to our new classification system presented in figure 2F, we were able to reduce ML of undefined use down to 36%. We describe this progress in line 319ff and specifically reference to fragments as part of this problem in line 369ff.

We think that the benefit of using FTIR would only provide limited assistance with source attribution. If we could determine that a fragment is of foamed Polystyrene, we would not know if it originates from food packaging (consumer related) or from insulation (industrial). We have used a NIR Spectrometer by Trinamix on site during monitoring in summer 2023, but wet, dirty objects, Polystyrene and dark objects were hard to be identified. FTIR could probably lead to better results in terms of polymer discrimination, but the resources are currently not available in our project and, as mentioned above, the benefit would be limited in our view.

Compatibility with International Standards: While adopting the MSFD list, the novel "source sector" and "durability" groupings are not yet widely standardized, potentially limiting comparability with global studies. The paper should clarify how these additions align with or enhance existing frameworks (e.g., UN Marine Litter Guidelines).

Thank you for discussing this important topic of harmonization of methods and monitoring standards. In our study, we wanted to make sure that we meet the state of the art, which is the MSFD in our understanding. It is newer than the UN Marine Litter Guidelines and offers a more comprehensive hierarchical litter category list that can be translated into other monitoring approaches, such as OSPAR.

Based on the MSFD, we extended the classification system to reduce the share undefined ML items (Figure 2E vs. 2F) and gain more information about responsibilities and sources to derive reduction measures. We do not understand our system as a new competitive system but as a first suggestion to extend the current MSFD ML list and support harmonization efforts. To make this point clearer, we have adjusted the sentence in line 466ff as follows: "Further extensions of guidelines, for example, by implementing our suggested use, source, and duration groups, could be a step towards more refined monitoring methods and mitigating ML pollution in rivers."

12. Highlight the EU's upcoming tethered cap regulation (Line 356) and estimate its potential impact using study data (e.g., reduction in plastic caps/lids, 4.5% of ML).

Thank you for pointing out this interesting aspect. We are currently analysing the data just for the bottle cap categories and the effectiveness of the EU Directive (EU) 2023/1060 in a separate manuscript. Since the currently revised manuscript only includes the sampling until the beginning of 2024, so before the Directive came into place, we only wanted to refer to the regulation being relevant in future studies, as seen in line 359ff.

13. The study region (Germany) has robust recycling systems (e.g., deposit-return for glass/plastic), but the paper does not assess how policy effectiveness (or gaps like illegal dumping, overflowing bins during floods, Figure 6A) influences ML composition (e.g., why glass bottles still account for 5.3% of ML despite regulations).

We agree that understanding the sources and entry paths of ML, such as overflowing bins or personal behaviour, would make it easier to derive and implement reduction measures. We do mention the possibilities of overflowing bins (line 696) or increased littering during outside activities in summer months (line 416ff). However, further analysis was not within the scope of this study and we currently do not have the data to make assumptions about the entry paths. For example, we cannot say if a glass bottle came from a cultural event, was littered at the river shore or was mobilized from overflowing bins. But what we can provide is a dataset that shows that glass bottles still make up a large share of ML despite the robust recycling system in Germany. This offers a great starting point for future studies to investigate entry paths for specific ML categories. We hope that you can appreciate our perspective and the rationale behind our choice.

14. The statistical year excluded extreme floods (e.g., 2021 Rhine flood), yet the discussion notes floods can mobilize "100x more ML than normal levels" (Line 652). The model uses a modest "rising discharge factor" (1.0424) to account for this, but without quantifying rare high-flow events' contribution to annual transport (e.g., sensitivity analysis with historical flood data is missing).

We agree that extreme floods may play an important role in ML mobilization and transportation. We have decided to implement a discharge factor, which is based on the average discharge of the last 30 years and thus, also include the extreme floods of 1993 and 1995 in Cologne, and not an extra flooding factor because of lacking data. The only available study on flooding events is from van Emmerik et al., 2023 and is based on visual counting from a bridge for several hours per day, one day per month. During the flood, the observations were made on one to three days, depending on location. The non-continuous measurements and few data points do not allow a high enough resolution to determine the influence of rising discharge during non-flood periods. For example, in Figure 1A and C of that publication, the mean plastic flux did not increase or there was no data available during a high increase of discharge at that time of the year. With our continuous approach, we could show that ML can vary by factor of 41 only based on normal annual rising and falling discharges. Additionally, the data points measured during the floods by van Emmerik et al., 2023 limit a good fit of linear or power function models to the data (Figure 2 in that publication). The linear model only fits better because it was split into two transport modes, which may indicate that the relation is likely non-linear given that more data points would be available. Without doubt, this publication is a very important first indicator that flooding events play an important role. But the data availability was not sufficient for us to reliably include it in our weighted model. We explain this in line 696ff. In the future, an upgraded design could be prepared to capture sampling data during extreme floods to close this data gap and improve extrapolation models.

Climate Change Omissions: Future increases in extreme precipitation in the Rhine basin may alter seasonal ML transport (e.g., more winter peaks), but the study does not engage with climate model projections or discuss long-term trend implications.

We agree that future changes due to climate change, for example the occurrence of rain events or floods as mentioned by Helen et. al., (2022), will have an influence on ML transport mechanisms in rivers. However, we think that developing climate model projections are outside the scope of this publication. It would be great to realize such a study in the future in collaboration with experts in this field of research.

Reviewer #2 (Remarks to the Author):

The paper provides data from a continuous 16-month monitoring of macro-litter using a litter trap in the Rhine River. The monitoring approach including participation of citizen scientist is explained in detail and appropriate for the task. The data assessment and statistics are well done, plausible and justified. A thorough discussion highlights the importance of continuous monitoring and compares the findings with those from other studies. The new data base for the amount of macro-litter transported by the Rhine River and disposed to the Northern Sea, the experiences gained using the litter trap and the well prepared discussion are a valuable contribution to science and to conclude on measures to minimize macro-litter emissions.

Dear Reviewer, thank you for your valuable feedback and taking the time to check the format and the supplement.

The structure of the manuscript has to be corrected, chapter 2 (Results) has got a wrong number and should be replaced by chapter "Methods" (line 433ff). Chapter 4 should be Discussion.

Based on recent publications, the order of chapters is correct, but the chapter titles are not numbered. We therefore removed all chapter numbers in the manuscript. Please see the following article that we used as reference for the Journal style: <https://www.nature.com/articles/s41893-025-01512-0>

Some minor comments: L48 "a rise in human carcinogenic toxicity" would need a more specific reference additionally to the related general reference [7].

Thank you for pointing this out. The OECD mentions carcinogenic toxicity but there is not much backup from scientific publications. We therefore change the text in line 48 to "... a rise in adverse effects on human health, ..." and additionally cite Yang et. al., 2022, Environmental health impacts of microplastics exposure on structural organization levels in the human body.

Fig. F: legend central pie chart: infrastructure .. construction

Thank you for catching this. We have fixed these mistakes.

L140: Please be precise and make clear that not sampling dates are compared but sampling periods of two weeks.

Thank you for pointing this out. Line 144 now reads: "...ML per biweekly sampling periods varies by..."

L216: 4 Discussion

Thank you for this hint. We deleted the chapter and subchapter numbers in the document.

L270: .. zinc, other heavy metals ..

Thank you. We have changed the text as suggested in line 300.

L308: A comment on potential risks from wooden corks may be added. As it is named together with other litter known to result in risks for nature etc., the potential impact of wooden corks is less obvious.

We agree with your statement. We have removed wooden corks from the sentence and replaced it with plastic sweet wrappers in Line 337f.

L502: Not clear what is $d = 1 \text{ g}$, Do you mean min 1 g?

This is a value that is given for the used digital scale and $d = 1 \text{ g}$ is correct. It indicates that the smallest increment this scale can display is 1 g. To make this more clear for the reader, we changed the scale details in Line 543f to: Soehnle, Page Compact 300, max 5 kg, display resolution of 1 g.

L693ff: Adjust format of reference according to journal guidelines, check ref. [10]-not listing all authors; [16]-add web-link; [18, 51]-vol/no of journal missing; The Science of The Total Environment

Thank you for this comment. We checked all the references you mentioned.

- Regarding reference [10], we checked the authors and could just find the five authors already mentioned in the reference list: Lourens J. J. Meijer, Tim van Emmerik, Ruud van der Ent, Christian Schmidt, Laurent Lebreton.
- Regarding the reference number [16] (doi 10.2760/127473), we gave this reference the type "report / grey literature", which is displayed differently and excludes the doi in the reference list compared with the literature in the type "journal article" based on the Nature reference style.
- Vol number 87 and the doi were added for reference [18].
- The reference [51] was referenced as pre-print version. As it was not published with a peer reviewed Journal yet, we excluded it from our manuscript (doi: [10.31223/X50Q45](https://doi.org/10.31223/X50Q45))

Supplementary Information:

L28: The litter trap (LT) ..

Thank you for catching that oversight.

L42, 43: add blank between numbers and units

Fixed, thank you.

L51 vs. L 54: 6 sampling dates or $N=5$?

Thank you for noticing that. The numbers were correct, but we added a bit more context. We did measure at six sampling dates, but we had to exclude one measurement for the LT entrance as the usual measuring point was blocked by a tree. We specified that in the supplement text in line 54ff.

L69: Fig. SI2.1

Fixed, thank you.

L72: (C: or (c) for copyright?

Fixed, thank you.

Reviewer #3 (Remarks to the Author):

The manuscript presents a thorough and innovative study on riverine macrolitter (ML) in the Rhine River, one of Europe's largest and most significant waterways. Over a 16-month period, the authors implemented a continuous monitoring strategy using a custom-designed physical-interception litter trap, which was maintained and operated in collaboration with local citizen scientists. This approach enabled the collection of 20,339 floating ML items larger than 1 cm, which were then classified using internationally recognized standards.

The dataset reveals that private consumers are the predominant source of ML, accounting for 56.4% of the total, with plastics comprising 69.7% of items by count but only 14.8% by weight. This highlights the significant role of non-plastic materials in the overall mass of ML. The authors also demonstrate that river discharge is a strong predictor of ML abundance, suggesting that hydrological conditions play a crucial role in the mobilization and transport of litter. Based on their data, the authors estimate that the Rhine transports between 27 and 42.2 million ML items (3,010.5 to 4,707.5 tons) annually toward the North Sea—figures that substantially exceed previous European estimates. The study documents high temporal variability in ML abundance, with biweekly sample counts fluctuating by a factor of 41, underscoring the importance of long-term and continuous monitoring to accurately capture the dynamics of riverine litter.

One of the manuscript's greatest strengths lies in its methodological rigor and innovation. The use of a custom-built litter trap capable of intercepting both floating and submerged ML up to 80 cm below the surface represents a significant advance over traditional visual observation methods, which are often limited by field conditions and observer bias. The long-term, biweekly sampling regime provides a robust dataset that captures both seasonal and event-driven variability. The integration of citizen science not only made extended monitoring feasible but also fostered local engagement and awareness. The comprehensive classification of ML by material, use, source, and sector—following the MSFD list and further refined by the authors—enhances the relevance of the findings for both scientific and policy audiences. The statistical analysis is sound, with clear evidence presented for the influence of environmental drivers such as discharge, precipitation, and wind speed on ML abundance and composition.

Despite these strengths, the study has some limitations. The mesh size of the litter trap precluded the systematic collection of ML smaller than 5 cm, which may lead to an underestimation of the contribution of smaller macrolitter and microplastics.

Additionally, a significant portion of the collected ML (about 30.6%) could not be definitively assigned to a source or use category, primarily due to the prevalence of fragments and ambiguous items, limiting the precision of source attribution.

The annual transport estimates are based on a single year that did not include extreme flooding events, so they may not fully capture interannual or episodic variability that could

significantly influence ML fluxes. The study's spatial scope is limited to a single site in Cologne, which, while strategically chosen, may not reflect spatial heterogeneity along the river's course.

Furthermore, aside from a clear spike in firework-related litter following New Year's Eve, the study was unable to link other cultural events to changes in ML abundance, possibly due to the overriding influence of hydrological factors or insufficient event-specific data.

The manuscript is clearly written and well-organized, with informative figures and tables that effectively support the narrative. The use of international classification standards and the detailed breakdown of ML types add clarity and facilitate comparisons with other studies. The findings have important policy implications—particularly the strong correlation between ML abundance and river discharge—which could inform targeted clean-up efforts and monitoring strategies. The study underscores the need for harmonized, long-term monitoring of riverine ML to support EU and international marine litter reduction goals.

To further strengthen the work, future studies could complement the current methodology with approaches targeting smaller ML and microplastics, expand sampling to additional sites along the river to assess spatial variability, and incorporate longer-term and event-based monitoring to refine annual flux estimates and better capture the impact of extreme events. Advanced forensic or chemical analysis methods could also be explored to improve the identification of ambiguous or fragmentary ML items.

Dear Reviewer, thank you very much for this very encouraging feedback and pointing out the parts that need improvement. Please see our point by point answers below.

Specific comments

1. The manuscript notes that the litter trap's mesh size prevented systematic sampling of ML items smaller than 5 cm, which represent 23% of the collected items but were not fully accounted for. Please provide a more detailed discussion on how this limitation may have influenced the representativeness of your results, particularly regarding the total ML flux and the composition of smaller litter classes. Consider suggesting how future studies might address this gap.

Thank you for this remark on this very important point that was also identified by reviewer 1. The mesh size influences our results on the ML numbers more than the ML weight since small items weight much less. Therefore, the data points in Figure 5A at 50 mm will probably deviate even more. We have now added an asterisk in the figure and expanded the figure caption to highlight that point. It now says in line 219ff: "The asterisk (*) indicates that this size class is smaller than the systematic sampling size and the calculated concentration for <50 mm is likely underestimated."

Figure 2 is based on ML numbers and is potentially also affected by an underestimate of small ML items like sweet wrappers or plastic lids of drinking bottles. We added text on line 289ff: "With a share of 69.7% by number, which presumably underestimates the share of small ML due to the mesh size of 4.24 cm, we also found plastics to be the most common material in the Rhine" and added "by number" to the Figure 2 caption in line 132.

Regarding the total ML flux, we have added a few sentences in line 257ff: "Regarding the size distribution, the total estimates are probably underestimating ML under 5 cm

as 4.24 cm is the lower limit of the systematic sampling size. Small ML below this size limit, such as fragments, cigarette butts, bottle caps, lids or toys, have a share of 23% of all ML by number (Figure 2C). Based on our finding that smaller ML is higher in number and lower in weight (Figure 5A), this underestimate of ML will more effect the extrapolation of total ML number than total ML weight transported in the Rhine.”

To solve this problem in the future, we wrote that it would be beneficial to add smaller filters to retain meso- and microlitter in line 456f. This would allow the ML items >2.5cm to be sampled systematically. However, this recommendation is conditional, since local conditions and the fauna in the water body must first be taken into account to avoid any health threats.

2. A significant proportion of the collected ML (about 30.6%) could not be assigned to a specific source or use group, mainly due to fragments and ambiguous items. Please elaborate on the challenges faced in source attribution and discuss whether additional analytical techniques (e.g., chemical fingerprinting, forensic analysis) could help improve the identification of these ambiguous items in future research.

Thank you for making up this point. It was also proposed by reviewer 1. We tried to decrease the share of 54% of undefined items (Figure 2E) based on the given classification system of the MSFD developed by Fleet et al. (2021) by using a new classification system presented here (Figure 2F). This lead us to still 36% of non defined items of which 30.6% were fragments. Therefore, an update of the MSFD ML list could mitigate that problem. Furthermore, the EU MSFD list includes categories which are titled as “other” like “other textiles”, “other glass items”, “other ceramic items”, “other identifiable foamed plastic items”, “other identifiable non-foamed plastic items”, “other paper items”, “other rubber pieces”, “other processed wooden items”. These categories with 5.4% include identifiable items which could not be precisely assigned to one of the other categories. Theoretically, we could use our photo catalogue to take another close look at each of these items (e.g. pacifiers are other rubber items) and assign them to the different usage groups we created. We will consider this option in our next publication to improve source attribution.

Regarding fragments, we believe that the benefit of using additional analytical techniques would be limited in helping source attribution. If we could determine that a fragment is of foamed Polystyrene, we would not know if it originates from food packaging (consumer related) or from insulation (industrial). We have used an NIR Spectrometer by Trinamix on site during monitoring in summer 2023, but wet, dirty objects, Polystyrene and dark objects were hard to be identified. Chemical fingerprinting or forensic analysis could probably lead to better results in terms of polymer discrimination, but the resources are currently not available in our project. If databases for chemical fingerprinting would allow a specific source attribution, then this could be beneficial in future studies but will still require a higher resource investment.

3. The annual ML transport estimates are based on a single year without extreme flooding events. Please discuss the potential variability in ML fluxes that might occur during years with a typical hydrological conditions, such as major floods or droughts. Clarify how representative your sampling year is for long-term trends and consider including a sensitivity analysis or uncertainty estimate.

Since reviewer 1 had a very similar comment, we copied our answer: We agree that extreme floods may play an important role in ML mobilization and transportation. We have decided to implement a discharge factor, which is based on the average discharge of the last 30 years and thus, also include the extreme floods of 1993 and 1995 in Cologne, and not an extra flooding factor because of lacking data. The only available study on flooding events is from van Emmerik et al., 2023 and is based on visual counting from a bridge for several hours per day, one day per month. During the flood, the observations were made on one to three days, depending on location. The non-continuous measurements and few data points do not allow a high enough resolution to determine the influence of rising discharge during non-flood periods. For example, in Figure 1A and C of that publication, the mean plastic flux did not increase or there was no data available during a high increase of discharge at that time of the year. With our continuous approach, we could show that ML can vary by factor of 41 only based on normal annual rising and falling discharges. Additionally, the data points measured during the floods by van Emmerik et al., 2023 limit a good fit of linear or power function models to the data (Figure 2 in that publication). The linear model only fits better because it was split into two transport modes, which may indicate that the relation is likely non-linear with more data points. Without doubt, this publication is a very important first indicator that flooding events play an important role. But the data availability was not sufficient for us to reliably include it in our weighted model. We explain this in line 696ff. In the future, we will be able to capture sampling data during extreme floods with the LT to close this data gap and improve extrapolation models.

4. The study is based on a single sampling site in Cologne. Please discuss the extent to which this location is representative of the entire Rhine River and whether local factors (e.g., urban density, tributary inputs, river engineering) may have influenced your findings. Suggest how additional sites along the river could enhance the spatial resolution and generalizability of the results.

Thank you for making this point. Additional sites with LTs for data collection along the Rhine would definitely increase the spatial resolution and generalizability of the results. For now we mainly look at the temporal resolution, as mentioned in line 447f. However, we expanded our discussion on the LT location for the entire Rhine river and added sewer outlets in line 444ff: "In addition, several LTs can be positioned at multiple locations along the river's length or cross-section to improve spatial resolution and evaluate local ML inputs, for example originating from cultural events or sewer outlets."

90% of the entire river network is located upstream the LT, 72% of the population live upstream the LT and this region covers agricultural as well as urban land. In addition to that, we assume to have a steady-state of ML within the water column (line 694f) due to continuous deposition-resuspension. Therefore, we consider the location of the LT at Cologne as representative for the whole river basin.

As part of our ongoing work, a second litter trap will be implemented, probably next year, in Wiesbaden to increase the spatial resolution. Other studies investigating ML along the river shores are planned in the future to gain more data for the spatial differentiation.

5. While a clear spike in firework-related litter was observed after New Year's Eve, the manuscript states that other cultural events could not be linked to changes in ML abundance.

Please clarify the methods used to identify and analyze the impact of cultural events. Consider whether more granular event-specific data or targeted sampling during known high-activity periods could provide further insights into the influence of human activities on riverine ML.

Yes, we agree that sampling at more locations with either more sampling dates around cultural events may improve the data to find a correlation. To plan accordingly, this study helps to set a baseline and better monitor cultural events in between the biweekly sampling dates.

In Germany, rockets, firecrackers and batteries can only be sold within the three days before New Years Eve and lighted on New Years Eve or the 1st of January. Due to this regulation a clear connection could be made for firework related ML. For all other events, we also used graphical analysis to identify peaks potentially related to events. However, there are two reasons why a relation of event and ML abundance was difficult to identify. On the one hand, the lack of a clear source attribution prevented identifying an impact of the events. For example, the sweet wrappers found could originate from Cologne Carnival or from people walking along the Rhine. On the other hand, rising discharge is a strong predictor of ML occurrence. We were not able to differentiate between the many factors overlaying a potential effect. For example, has the sweet wrapper been transported by rising discharge events over a longer distance or from a local event?